

# Effective free-fermionic form factors and the XY spin chain

O. Gamayun[1,2★], N. Iorgov[1,3] and Yu. Zhuravlev[1]

**1** Bogolyubov Institute for Theoretical Physics, 03143 Kyiv, Ukraine
**2** Institute for Theoretical Physics, University of Amsterdam,
1090 GL Amsterdam, The Netherlands
**3** Kyiv Academic University, 03142 Kyiv, Ukraine

★ oleksandr.gamayun@gmail.com

## Abstract

We introduce effective form factors for one-dimensional lattice fermions with arbitrary phase shifts. We study tau functions defined as series of these form factors. On the one hand we perform the exact summation and present tau functions as Fredholm determinants in the thermodynamic limit. On the other hand simple expressions of form factors allow us to present the corresponding series as integrals of elementary functions. Using this approach we re-derive the asymptotics of static correlation functions of the XY quantum chain at finite temperature.



# 1  Introduction

Exactly solvable one-dimensional quantum mechanical systems of interacting spins, bosons, and fermions provide a unique platform for studying non-perturbative effects. The algebraic and coordinate Bethe ansatz allow one to find the wave functions [1] and analytically address the thermodynamic properties of these systems [2]. The matrix elements of physical operators can also be found analytically in many cases [3–6], but the computation of correlation functions still remains quite challenging. For the vacuum correlation functions there are effective numerical methods based on integrability [7]. The asymptotic behaviour of the correlation functions can be investigated by means of effective field theory (Luttinger liquid) [8]. The origin of this behavior has been linked to the finite-size scaling of the matrix elements computed by means of the Bethe Ansatz [9–13]. For dynamical correlation functions based on this approach the corresponding effective field theory bears the name of *non-linear* Luttinger liquid [14–16].

At finite temperature, or more generally at finite entropy (density of states), both the numerical and field theory approaches experience some difficulties. In the numerical approaches one has to scan a much larger portion of Hilbert space to saturate the sum rules, as the form-factors (matrix elements of the physical operators) decay exponentially with the systems size contrary to the power-law decays at zero temperatures (see for instance [17, 18]). The field theory approach is based mainly on the linear spectrum for the soft modes (low-energy excitations) which is valid only for very low temperatures [19, 20].

A more rigorous approach was developed to evaluate finite temperature correlation function in integrable lattice models of Yang–Baxter type, based on the Quantum Transfer Matrix (QTM) [21]. The notion of the thermal form factor was introduced [22], which turned out to be useful for the asymptotic analysis of two-point functions [22–24]. In the scaling limit, thermal form factors also arise axiomatically in the context of Integrable Quantum Field Theory [25–33]. Less rigorous but numerically accurate approaches are based on the thermodynamic limit of the form-factors and restricting summation to a finite number of particle-hole pairs [34–37].

However, the complete understanding has not been achieved and recently the QTM methods were revisited to address correlation function for the XX spin-chain [38–40]. Moreover, new systematic approaches for correlation functions in the Ising model for low density [41] and in the Lieb–Liniger model for the strong-coupling expansions [42] have been proposed. The generalization of Smirnov's form factor axioms for the thermodynamic states has been formulated in Ref. [43] and successfully applied to the reconstruction of the generalized hydrodynamic description of the correlation functions [44, 45].

In this work we develop a heuristic approach to address asymptotics of correlation functions at finite density of states. Our main motivating example is the XY spin chain in a transverse magnetic field. On one hand these systems were analyzed extensively in literature and the exact answers for spin-spin correlation functions in terms of Toeplitz or Fredholm determinants are known. On the other hand the complexity of excitation is the same as for generic systems. As we have mentioned above, this complexity is combinatorial in nature and reflects the fact that each form factor for the thermal states is exponentially small so the number of relevant form factors is exponentially large. This makes direct computation of the corresponding sum for the correlation functions notoriously difficult and force researchers to focus at most on the two particle-hole excitations [34–37], consider semiclassical approximations [46] or develop other approximation schemes [41, 42].

We deal with this problem in a different manner. Namely, to describe the spin-spin correlation function evaluated on a state with finite density of entropy (energy) we introduce *effective* form factors for the fermions with the modified phase shift that absorbs information about the state and significantly simplifies combinatorics of excitations making it essentially analogous to the zero-temperature case. Here we have to emphasize that the expressions of form factors was inspired by the XX spin chain [47–50], rather than genuine spin form factors in the Ising/XY models [51–55].

We focus on the static correlation functions for which we demonstrate that after complete summation of the effective form factors series and taking the thermodynamic limit the answer can be presented in the form of Fredholm determinants. The method of form factors summation of this type was pioneered in Ref. [56] for the correlation functions in the impenetrable Bose gas model (see also [57]). For the proper choice of the phase shift in the effective form factors the kernels in the Fredholm determinants differ from the exact ones [50] by the exponentially small (in distance between spin operators) terms. Conversely, by first taking the thermodynamic limit of the effective form factors and then performing their summation we manage to present the Fredholm determinants as integrals of elementary functions. This kind of asymptotic behavior for models in the continuum (not the lattice) arises similarly from the solution of the Riemann–Hilbert problem for operators acting on the whole real line [58]. This asymptotics was conjectured to be universal for correlation functions of any gapless model of statistical mechanics at any temperature and for an arbitrary coupling constant [59].

An important ingredient for our asymptotic analysis is the winding number of the state-dependent phase shift $\nu(q)$ defined as the difference across the Brillouin zone, namely

$$\nu(+\pi) - \nu(-\pi) = \delta \in \mathbb{Z}. \tag{1}$$

We recover the correlation length in the lattice version of the asymptotics in Ref. [59] at $\delta = 1$ and additionally give an analytic expression for the prefactor. For $\delta = 0$ and $\delta = \pm 1$ we derive asymptotic behavior for the correlation function in the XY spin chain at finite temperature and compare it with the known answer [60]. Different winding numbers correspond to different values of the magnetic field and anisotropy. The winding number $|\delta| \geq 2$ does not have a direct physical interpretation in this model, but we perform the asymptotic analysis anyway and find the results consistent with the generalization of Szegő formulas [61]. Moreover, we have observed a peculiar identity between Toeplitz determinants and Fredholm determinant of sine-kernel type with finite rank, which, to the best of our knowledge, is new

$$\det\left(1 + \hat{S}_\nu + \delta \hat{V}_\nu\right) - \det\left(1 + \hat{S}_\nu\right) = \det_{1 \leq j,k \leq x} T_{j-k}, \tag{2}$$

where the operators $\hat{S}_\nu$ and $\delta \hat{V}_\nu$ are generalized sine-kernels that act on $L^2([-\pi, \pi])$ and are defined by their kernels

$$S_\nu(p,q) = \frac{e^{2\pi i \nu(p)} - 1}{2\pi} \frac{\sin \frac{x(p-q)}{2}}{\sin \frac{p-q}{2}}, \qquad \delta V_\nu(p,q) = -\frac{e^{2\pi i \nu(p)} - 1}{2\pi} e^{-ix(p+q)/2} e^{-i(p-q)/2}, \tag{3}$$

and

$$T_k = -\frac{1}{2\pi} \int\limits_{-\pi}^{\pi} d\varphi \, e^{-i(k+1)\varphi + 2\pi i \, \nu(\varphi)}. \tag{4}$$

Notice that the right hand side of Eq. (2) can be also be presented as a Fredholm determinant but with the modified shifted $\nu(k)$ [62]. If $\nu(k)$ corresponds to the XY spin chain the explicit Fredholm determinants are given in Eq. (74). For the same $\nu(k)$ the left hand side of Eq. (2) was obtained in [50] and the right hand side in [60, 63] (as Toeplitz determinant).

The paper is organized as follows. In Sec. 2 we define the tau function together with the effective form factors and outline the derivation of the Fredholm determinant presentation resulting from the summation of form factors. The details of this derivation are presented in Appendix A. In Sec. 3 we study the thermodynamic limit of the form factors and argue for an explicit presentation of the form factors series as integrals of elementary functions. All necessary technical results are given in Appendices B and C. Sec. 4 deals with the application of the general formulas to the XY spin chain. In Sec. 5 we discuss connection of the general result to the Toeplitz determinant and relations such as Eq. (2). Sec. 6 concludes the paper and offers an outlook.

## 2 Effective form factors

We start with the formal definition of the static correlation function (tau function), as a form-factor series

$$\tau(x) \equiv \sum_{\mathbf{q}} |\langle \mathbf{k}|\mathbf{q}\rangle|^2 e^{-ix\left(\sum\limits_{i=1}^{N+1} k_i - \sum\limits_{i=1}^{N} q_i\right)}, \tag{5}$$

here the ordered set $\mathbf{k} = \{k_1, \ldots, k_{N+1}\}$ consists of $N+1$ distinct shifted momenta inside the Brillouin zone ($k_i \sim k_i + 2\pi n$, $n \in \mathbb{Z}$) each being a solution of the transcendental equation

$$e^{ikL} = e^{-2\pi i \, \nu(k)} \tag{6}$$

for a smooth function $\nu(k)$. This function plays the role of the phase shift and is assumed to be compatible with the Brillouin zone structure, i.e.

$$\nu(\pi) - \nu(-\pi) = \delta \in \mathbb{Z}. \tag{7}$$

The integer $\delta$ is the winding number (index). One can easily argue that the number of solutions of Eq. (6) is $L + \delta$, each root defined up to $O(1/L^2)$ terms.

The set $\mathbf{q} = \{q_1, \ldots, q_N\}$ is an ordered set of $N$ distinct solutions of equation

$$e^{iqL} = 1. \tag{8}$$

Further we consider different values of $N$ and $L$, provided that the sets $\mathbf{q}$ and $\mathbf{k}$ are not empty. For given sets, motivated by the spin form factors for quantum XY chain written in the XXO basis [50], we postulate the following form-factor

$$|\langle \mathbf{k}|\mathbf{q}\rangle|^2 = -\frac{4L}{\prod\limits_{i=1}^{N+1}(1 + \frac{2\pi}{L}\nu'(k_i))} \left(\prod_{i=1}^{N+1} \frac{e^{g(k_i)/2}\sin\pi\nu_i}{L}\right)^2 \prod_{i=1}^{N} e^{-g(q_i)}(\det D)^2, \tag{9}$$

where $\det D$ is a trigonometric Cauchy type determinant that can be presented in two equivalent forms

$$\det D = \begin{vmatrix} \cot\frac{k_1-q_1}{2} & \dots & \cot\frac{k_{N+1}-q_1}{2} \\ \vdots & \ddots & \vdots \\ \cot\frac{k_1-q_N}{2} & \dots & \cot\frac{k_{N+1}-q_N}{2} \\ 1 & \dots & 1 \end{vmatrix} = \frac{\prod\limits_{i>j}^{N+1}\sin\frac{k_i-k_j}{2}\prod\limits_{i>j}^{N}\sin\frac{q_j-q_i}{2}}{\prod\limits_{i=1}^{N+1}\prod\limits_{j=1}^{N}\sin\frac{k_i-q_j}{2}}. \tag{10}$$

Furthermore, since we do not specify the specific operator we will sometimes refer to Eq. (9) as to the overlap, and use this term interchangeably with form factor.

We assume that the index is of order $\delta \sim O(1)$ as both the system size and the number of particles are approaching the thermodynamic limit $N \to \infty$, $L \to \infty$ such that $N/L = 1$. In this case, the summation over $\mathbf{q}$ can be performed exactly, similarly to Ref. [56] (see Appendix A). The result for the tau function reads

$$\tau(x) \stackrel{N\to\infty}{=} \det(1+\hat{V}+\delta\hat{V}) - \det(1+\hat{V}), \tag{11}$$

where the determinants are taken in the space $L^2(S^1)$ and the corresponding operators are defined by their kernels

$$V(k,q) = \frac{\sin^2(\pi\nu(k))}{4\pi}e^{g(k)}e^{-i(k+q)x/2}e^{i(k-q)/2}\frac{E(k)-E(q)}{\sin\frac{k-q}{2}}, \tag{12}$$

$$\delta V(k,q) = -\frac{2}{\pi}\sin^2(\pi\nu(k))e^{g(k)}e^{-i(k+q)x/2}, \qquad k,q \in [-\pi,\pi) \tag{13}$$

with

$$E(k) = \int\limits_{-\pi}^{\pi}\frac{dq}{\pi}e^{-g(q)+iqx}\cot\frac{q+i0-k}{2} - \frac{4ie^{-g(k)+ikx}}{e^{-2\pi i\nu(k)}-1}. \tag{14}$$

The diagonal terms $k = q$ are understood as in L'Hopital's limiting procedure. Further, we impose the relation

$$e^{-g(k)} = e^{-2\pi i\nu(k)} - 1 \tag{15}$$

to present tau function as

$$\tau(x) = \det(1+\hat{S}_\nu+\delta\hat{V}_\nu+\hat{R}) - \det(1+\hat{S}_\nu+\hat{R}), \tag{16}$$

with $\hat{S}_\nu$ being a generalized sine-kernel

$$S_\nu(k,q) = \frac{e^{2\pi i\nu(k)}-1}{2\pi}e^{i(k-q)/2}\frac{\sin\frac{x(k-q)}{2}}{\sin\frac{k-q}{2}}, \qquad \delta V_\nu(k,q) = -\frac{e^{2\pi i\nu(k)}-1}{2\pi}e^{-ix(k+q)/2}. \tag{17}$$

This way, the remainder $\hat{R} = \hat{V} - \hat{S}_\nu$ consists of integrals in Eq. (14), which are exponentially suppressed[1] for large and positive $x$. Let us call the tau function with discarded $\hat{R}$ as $\tau_S$, namely

$$\tau_S(x) = \det(1+\hat{S}_\nu+\delta\hat{V}_\nu) - \det(1+\hat{S}_\nu). \tag{18}$$

This particular generalization of the sine-kernel is contained in the prefactor $(e^{2\pi i\nu(k)}-1)$ and allows one to describe a modification of the system from the vacuum state for which $\nu(q)$ is constant within the arc $k \in [-k_F, k_F]$ and zero everywhere else, to the finite-entropy state, where, for instance, for the thermal state of the fermionic system the prefactor would be proportional to the single-particle Fermi distribution function[2]. In Sec. 4 we relate this type of kernel to the static spin-spin correlations in the XY chain. Then $\nu(k)$ will depend not only on the state but also on the parameters of the model.

---

[1] We assume that $\exp(-2\pi i\nu(q))$ is an analytic function within some vicinity of the real line.

[2] See, for instance the discussion in Appendix A in Ref. [64].

# 3 Thermodynamic limit and direct summation of form factors

## 3.1 Winding number $\delta = 1$

In the previous section, we considered the summation of the form factor series and subsequent taking of the thermodynamic limit. This leads to the presentation of the tau function as a Fredholm determinant. The essence of this derivation, which is outlined in Appendix A, is that each momentum $q_i \in \mathbf{q}$ was treated independently. In this section, we focus more on the detailed structure of the ordered sets $\mathbf{q}$ in the sum of Eq. (5). The total number of solutions of the equation $e^{iqL} = 1$ inside the Brillouin zone is $L$, which can be presented as

$$q_j = \frac{2\pi}{L}\left(-\frac{L+1}{2} + j\right), \qquad j = 1, 2, \ldots, L. \tag{19}$$

As we have already pointed out above, the number of solution of Eq. (6), depends on the winding number $\delta$. In particular, for $\delta = 1$ there exist exactly $L + 1$ solutions inside the Brillouin zone

$$k_j = \frac{2\pi}{L}\left(-\frac{L+1}{2} + j - v_j\right), \qquad v_j = v(k_j) \approx v(q_j), \qquad j = 1, 2, \ldots, L+1. \tag{20}$$

If we choose the set $\mathbf{k} = \{k_1 \ldots, k_{L+1}\}$ in Eq. (5) then summation over $\mathbf{q}$ will only involve one term $\mathbf{q} = \{q_1, \ldots q_L\}$. In the large $L$ limit the corresponding overlap reduces to a constant which is slightly counterintuitive from the orthogonality catastrophe point of view [65]. The explicit value of this constant is given by Eq. (213). The difference of momenta in Eq. (5) can be evaluated in the large $L$ limit as

$$\Delta P \equiv \sum_{i=1}^{L+1} k_i - \sum_{i=1}^{L} q_i \approx \pi - \int_{-\pi}^{\pi} v(q)dq. \tag{21}$$

Combining these observations together we obtain explicit equality for the Fredholm determinants in Eq. (11), and approximation for the generalized sine-kernel

$$\tau_S(x) \approx \tau(x) = \exp\left(-i\pi x + ix\int_{-\pi}^{\pi} v(q)dq - \frac{1}{2}\int_{-\pi}^{\pi} dq \int_{-\pi}^{\pi} dk \left[\frac{v(q) - v(k) - (q-k)/2\pi}{2\sin\frac{q-k}{2}}\right]^2\right). \tag{22}$$

It is interesting to note that only the periodic (i.e. having winding number $\delta = 0$) part of $v(q)$ has entered the final answer.

## 3.2 Winding number $\delta = 0$

For $\delta = 0$ we proceed similarly to the previous subsection. This time however the maximal possible number of the $k_i \in \mathbf{k}$ is $L$, so the maximal set $\mathbf{q}$ consists of $N = L - 1$ momenta. There are exactly $L$ such sets and they can be parameterized by the position of the "hole"

$$\mathbf{q}^{(a)} = \{q_1, \ldots, q_{a-1}, q_{a+1}, \ldots, q_L\}, \qquad a = 1, 2, \ldots, L. \tag{23}$$

The overlap is given by

$$|\langle \mathbf{k} | \mathbf{q}^{(a)} \rangle|^2 = \frac{A[q_a]e^{g(q_a)}}{L}\left[\frac{\Gamma(L - a + 1 - v_a)\Gamma(a + v_a)}{\Gamma(L - a + 1 - v_+)\Gamma(a + v_+)}\right]^2 \left(\frac{\pi + q_a}{\pi - q_a}\right)^{2v_+ - 2v_a}. \tag{24}$$

The derivation can be found in Appendix C.3 along with the explicit expression for $A[q_a]$ (see Eq. (227)). For $a \sim L$ and $L - a \sim L$ the last two factors cancel each other, which yields the following explicit expression

$$|\langle \mathbf{k} | \mathbf{q}^{(a)} \rangle|^2 = \frac{e^{-2\pi i \nu_a} - 1}{L} \exp\left(-\frac{1}{2} \int_{-\pi}^{\pi} dq \int_{-\pi}^{\pi} dk \left[ \frac{\nu(q) - \nu(k)}{2 \sin \frac{q-k}{2}} \right]^2 - \int_{-\pi}^{\pi} \nu(q) \cot \frac{q - q_a + i0}{2} dq \right). \tag{25}$$

On a technical side, we have used a variation of Sokhotski–Plemelj formula

$$\int_{-\pi}^{\pi} \nu(q) \cot \frac{q-k}{2} dq = \int_{-\pi}^{\pi} \nu(q) \cot \frac{q - k + i0}{2} dq + 2\pi i \, \nu(k), \tag{26}$$

to transform the integral in the exponential. For $a \sim 1$ and $L - a \sim 1$, the overlap is still $O(1/L)$, so we can replace the sum in tau function Eq. (5) by an integral

$$\tau(x) = e^{-ix \sum_{j=1}^{L}(k_j - q_j)} \sum_{a=1}^{L} |\langle \mathbf{k} | \mathbf{q}^{(a)} \rangle|^2 e^{-ix q_a} = T_0(x) Y_0(x), \tag{27}$$

with

$$T_0(x) = \exp\left( ix \int_{-\pi}^{\pi} \nu(q) dq - \frac{1}{2} \int_{-\pi}^{\pi} dq \int_{-\pi}^{\pi} dk \left[ \frac{\nu(q) - \nu(k)}{2 \sin \frac{q-k}{2}} \right]^2 \right), \tag{28}$$

and

$$Y_0(x) = \int_{-\pi}^{\pi} \frac{dk}{2\pi} (e^{-2\pi i \nu(k)} - 1) e^{-ikx} \exp\left( -\int_{-\pi}^{\pi} \nu(q) \cot \frac{q - k + i0}{2} dq \right). \tag{29}$$

Equivalently we may re-write $Y_0(x)$ as a contour integral in the variable $z = e^{ik}$

$$Y_0(x) = \frac{1}{2\pi i} \oint_{C_>} \frac{dz}{z} (e^{-2\pi i \nu(k)} - 1) z^{-x} \mathfrak{S}(z), \tag{30}$$

where the contour $C_>$ is a circle centered at the origin with slightly larger than unit radius and

$$\mathfrak{S}(z) = \exp\left( i \int_{-\pi}^{\pi} dq \, \nu(q) \frac{z + e^{iq}}{z - e^{iq}} \right). \tag{31}$$

We assume that $\nu(q)$ is non-singular in the region of integration, thus $\mathfrak{S}(z)$ is holomorphic outside the unit circle on the Riemann sphere, so the asymptotic for large positive integers $x$ is defined by the analytic behavior of $\nu(k)$ in the upper-half plane. For example, if $e^{-2\pi i \nu(k)}$ is a meromorphic function (of $z = e^{ik}$) outside the unit circle in the complex plane having simple poles at $z_1, z_2, \ldots$, with $1 < |z_1| < |z_2| < \cdots$, then for large $x$ the leading contribution comes from the smallest pole

$$Y_0(x) \approx -z_1^{-x-1} \mathfrak{S}(z_1) \operatorname{res}_{z=z_1} e^{-2\pi i \nu(k)}, \qquad z = e^{ik}. \tag{32}$$

Applying this formula together with Eq. (28), we have an asymptotic expression for the sine-kernel Fredholm determinant for $\delta = 0$

$$\tau_S(x) \approx \tau(x) \approx -\frac{T_0(x)}{z_1^{x+1}} \mathfrak{S}(z_1) \operatorname{res}_{z=z_1} e^{-2\pi i \nu(k)}. \tag{33}$$

**Remark**. Notice that even for $\delta = 1$ one could have chosen $\mathbf{k} = \{k_1, \ldots k_L\}$. This would not affect the derivation of the Fredholm determinants, but instead of one term in the form factor series as in the previous section, we still get a sum of $L$ terms. Using Appendix C.3 and specifically Eq. (229), we obtain

$$
\tau(x) = e^{-ix \sum_{j=1}^{L}(k_j - q_j)} \sum_{a=1}^{L} |\langle \mathbf{k} | \mathbf{q}^{(a)} \rangle|^2 e^{-ixq_a}
$$

$$
= \tau_{\delta=1}(x) \sum_{a=1}^{L} e^{ix\pi - ixq_a} \frac{\sin^2(\pi \nu_a)}{\pi^2} \frac{e^{2F(q_a)+g(q_a)}}{e^{2F(\pi)+g(\pi)}} \left[ \frac{\Gamma(L-a+1-\nu_a)\Gamma(a+\nu_a)}{\Gamma(L-a+2-\nu_+)\Gamma(a+\nu_+)} \right]^2. \quad (34)
$$

Here, by $\tau_{\delta=1}(x)$ we mean the r.h.s of Eq. (22). Notice that contrary to the $\delta = 0$ scenario, the middle parts $a \sim L$ and $L - a \sim L$, are suppressed as $1/L^2$, so their contributions are negligible as $L \to \infty$. The soft-modes at the edges $a \ll L$ and $L - a \ll L$ now start to play more important role because the corresponding overlaps are $O(1)$. The prefactor in front of the Gamma functions simplifies to one and the whole series reads

$$
\tau(x) = \tau_{\delta=1}(x) \frac{\sin^2(\pi \nu_-)}{\pi^2} \sum_{a=1}^{L} \left[ \frac{\Gamma(L-a+1-\nu_a)\Gamma(a+\nu_a)}{\Gamma(L-a+2-\nu_+)\Gamma(a+\nu_+)} \right]^2 + O(1/L). \quad (35)
$$

In order to compute this sum in the $L \to \infty$ limit we expand it at the edges and then perform the summation of the simplified expression extending the upper limit to infinity. Namely, the asymptotics

$$
\frac{\Gamma(L-a+1-\nu_a)\Gamma(a+\nu_a)}{\Gamma(L-a+2-\nu_+)\Gamma(a+\nu_+)} = \begin{cases} (a+\nu_-)^{-2} & , a \ll L \\ (L-a-\nu_-)^{-2} & , L-a \ll L \end{cases}, \quad (36)
$$

leads to

$$
\frac{\tau(x)}{\tau_{\delta=1}(x)} = \frac{\sin^2(\pi \nu_-)}{\pi^2} \left( \sum_{a=1}^{\infty} \frac{1}{(a+\nu_-)^2} + \sum_{a=0}^{\infty} \frac{1}{(a-\nu_-)^2} \right) = \frac{\sin^2(\pi \nu_-)}{\pi^2} \sum_{a=-\infty}^{\infty} \frac{1}{(a+\nu_-)^2} = 1. \quad (37)
$$

This way we restore the correct result even in the different formulation of the form-factor series.

### 3.3 Negative winding number $\delta < 0$

Let us consider $\delta = 1 - n$ for positive integers $n \in \mathbb{Z}_>$. The maximal number of solutions of Eq. (6) is $\ell = L + \delta$. We choose all of them to comprise our set $\mathbf{k}$

$$
\mathbf{k} = \{k_1, \ldots k_\ell\}, \qquad k_i = \frac{2\pi}{L}\left( -\frac{L+1}{2} + i - \nu_i \right). \quad (38)
$$

The set $\mathbf{q}^{a_1,\ldots a_n}$ is obtained from the complete set $\mathbf{q}$ in Eq. (19) by the omission of the "particle" (creating a "hole") at positions $q_{a_i}$

$$
\mathbf{q}^{a_1,\ldots a_n} = \{q_1, \ldots \hat{q}_{a_1}, \ldots \hat{q}_{a_n}, \ldots q_L\}. \quad (39)
$$

The total difference of momenta for such a state reads

$$
\Delta P_{a_1,\ldots a_n} = \sum_{i=1}^{\ell} k_i - \sum_{i=1}^{L} q_i + \sum_{i=1}^{n} q_{a_i} \approx \delta \pi - \int_{-\pi}^{\pi} \nu(q) dq + \sum_{i=1}^{n} q_{a_i}. \quad (40)
$$

The corresponding overlap is analyzed thoroughly in Appendix C.4 for $a_i \sim L$, $L - a_i \sim L$. It gives the following contribution to the tau function (5)

$$
e^{-ix\Delta P_{a_1,\ldots a_n}} |\langle \mathbf{k} | \mathbf{q}^{a_1,\ldots a_n} \rangle|^2 = \mathcal{A}_\delta[\nu] \prod_{i>j}^{n} \left( 2\sin \frac{q_{a_i} - q_{a_j}}{2} \right)^2 \prod_{i=1}^{n} \mathcal{Y}_{a_i}, \tag{41}
$$

where

$$
\mathcal{A}_\delta[\nu] = \exp\left( ix \int_{-\pi}^{\pi} \nu(q) dq - ix\delta\pi - \frac{1}{2} \int_{-\pi}^{\pi} dq \int_{-\pi}^{\pi} dk \left[ \frac{\nu(q) - \nu(k) - (q-k)\delta/(2\pi)}{2\sin\frac{q-k}{2}} \right]^2 \right) \tag{42}
$$

and

$$
\mathcal{Y}_a = -4 \frac{\sin^2(\pi\nu(q_a))}{L} \exp\left[ -ixq_a + g(q_a) - \int_{-\pi}^{\pi} dq \left( \nu(q) - \delta\frac{q}{2\pi} \right) \cot\frac{q - q_a}{2} \right]. \tag{43}
$$

In order to evaluate the tau function we proceed similarly to Ref. [66,67] (see also [68]). First, we notice that one can present the product of sines in (41) as Vandermonde determinants

$$
\prod_{i>j}^{n} \left( 2\sin \frac{q_{a_i} - q_{a_j}}{2} \right)^2 = \prod_{k>j}^{n} \left( e^{iq_{a_k}} - e^{iq_{a_j}} \right) \left( e^{-iq_{a_k}} - e^{-iq_{a_j}} \right)
$$

$$
= \det_{1\le j,k\le n} (e^{i(j-1)q_{a_k}}) \det_{1\le j,k\le n} (e^{-i(j-1)q_{a_k}}) = \varepsilon_{j_1\ldots j_n} \varepsilon_{j_1'\ldots j_n'} e^{i(j_1 - j_1')q_{a_1}} \ldots e^{i(j_n - j_n')q_{a_n}}, \tag{44}
$$

where $\varepsilon_{j_1\ldots j_n}$ is a completely antisymmetric tensor; the summation over repeated indices is implied. This expression is an almost factorized, so in the second step we render summation over $q_{a_i}$ to be independent, namely

$$
\sum_{q_{a_1} < \cdots < q_{q_n}} = \frac{1}{n!} \sum_{q_{a_1}} \cdots \sum_{q_{a_n}}. \tag{45}
$$

This immediately allows us to write the tau function (5) in the thermodynamic limit

$$
\tau(x) = \det_{1\le j,k\le n} [Y_\delta(x + j - k)] \exp\left( ix \int_{-\pi}^{\pi} \nu_\delta(q) dq - \frac{1}{2} \int_{-\pi}^{\pi} dq \int_{-\pi}^{\pi} dk \left[ \frac{\nu_\delta(q) - \nu_\delta(k)}{2\sin\frac{q-k}{2}} \right]^2 \right), \tag{46}
$$

where $\nu_\delta(q) \equiv \nu(q) - (q+\pi)\delta/(2\pi)$ has zero winding number and $Y_\delta(x)$ stands for

$$
Y_\delta(x) = \int_{-\pi}^{\pi} \frac{dq}{2\pi} \left( e^{-2\pi i\nu(q)} - 1 \right) \exp\left( -i(x-\delta)q + i\delta\pi - \int_{-\pi}^{\pi} dk\, \nu_\delta(k) \cot\frac{q - k + i0}{2} \right). \tag{47}
$$

The integral has been transformed using identities such as Eq. (26) to facilitate finding asymptotic behavior at large positive $x$. Indeed, the exponential is an analytic function, so after the proper deformation of the integration contour it can be dropped. This way, we demonstrate that, in fact, $Y_\delta(x)$ depends only on $\nu_\delta(x)$, namely

$$
Y_\delta(x) = \int_{-\pi}^{\pi} \frac{dq}{2\pi} e^{-2\pi i\nu_\delta(q)} \exp\left( -ixq - \int_{-\pi}^{\pi} dk\, \nu_\delta(k) \cot\frac{q - k + i0}{2} \right). \tag{48}
$$

Let us emphasize that Eq. (46) gives the exact answer for the Fredholm determinants (11). The asymptotic behavior for large $x$ will give also asymptotics for the generalized sine-kernel determinants (18). Similar to the treatment of the $\delta = 0$, the asymptotic expansion of $Y_\delta(x)$ is connected with analytic properties of $v(q)$. Let us assume that the first $n$ leading terms are given by

$$Y_\delta(x) = A_1 e^{-\varkappa_1 x} + \cdots + A_n e^{-\varkappa_n x} + o(e^{-\varkappa_n x}), \qquad |\varkappa_i| \le |\varkappa_{i+1}|. \tag{49}$$

Then the leading order of the determinant reads

$$\det_{1 \le j,k \le n} [Y_\delta(x+j-k)] \approx \prod_{i=1}^{n} A_i e^{-\varkappa_i x} \prod_{i>j}^{n} \left( 2\sinh \frac{\varkappa_i - \varkappa_j}{2} \right)^2. \tag{50}$$

For $n = 1$ we reproduce the results from the previous subsection.

## 3.4  Positive winding numbers $\delta > 1$

For $\delta > 1$, similar to $\delta = 1$, we can keep the maximal available number of $k_i \in \mathbf{k}$, so the r.h.s sum in Eq. (5) consists of one term, which is of order $O(1/L)$. This means that the corresponding Fredholm determinants in Eq. (11) vanish identically. The reason for this can already be seen before going into the thermodynamic limit. Namely, first we notice that the matrix $\mathcal{A}_{ij}$ in Eq. (129) can be considered on the full set of momenta $\mathbf{k} = \{k_1, \ldots, k_{L+\delta}\}$, which will not change the determinant's limiting value

$$\lim_{L \to \infty} \det_{1 \le i,j \le L+\delta} \mathcal{A}_{ij} = \det(1 + \hat{V}). \tag{51}$$

But since $\mathcal{A}_{ij}$ has the form of Eq. (127) which can be schematically written as

$$\mathcal{A}_{ij} = \sum_{k=1}^{L} \varphi_{q_k}(k_i) \phi_{q_k}(k_j), \tag{52}$$

for some functions $\varphi$ and $\phi$. This means that the rank of this matrix is maximally $L$ and addition of the rank one matrix $\delta V$ can increase the rank to at most $L + 1$. Therefore, for $\delta > 1$

$$\det(1 + \hat{V}) = \det(1 + \hat{V} + \delta \hat{V}) = 0. \tag{53}$$

The corresponding determinants with sine-kernels are not zero i.e. $\det(1 + \hat{S}_v) \ne 0$. In this case we see that even though the difference between $\hat{V}$ and $\hat{S}_v$ is exponentially small, it cannot be neglected, contrary to the cases for $\delta \le 1$. To estimate the difference between the Fredholm determinants with two different trace class operators one can use Eqs. (4.1) and (4.2) in [69]. These estimates in our case are in fact too rough as they also exclude discarding terms for $\delta \le 1$, even though numerically we can check that this operation is still legit. We found the following rule of thumb to neglect the exponential corrections for the kernel: the correction should vanish faster than the resulting determinant. For $\delta > 1$ this rule is violated, so to find the asymptotics we modify the definition of the tau function by considering summation over $\mathbf{k}$ in Eq. (5) instead of $\mathbf{q}$, namely

$$\tau_-(x) = \sum_{\mathbf{k}} |\langle \mathbf{k}|\mathbf{q}\rangle|^2 e^{-ix\left(\sum_{i=1}^{N+1} k_i - \sum_{i=1}^{N} q_i\right)}, \tag{54}$$

where the overlaps keep their form (9) but with the modified relation between $v(q)$ and $g(q)$, namely

$$e^{-g(q)} = e^{2\pi i v(q)} - 1. \tag{55}$$

In the thermodynamic limit this sum transforms into Fredholm determinants (see Appendix A)

$$\tau_-(x) = \det(1 + \hat{V}_- + \delta\hat{V}_-) + (\Gamma - 1)\det(1 + \hat{V}_-), \tag{56}$$

with

$$V_-(k,q) = \frac{e^{2\pi i \nu(k)} - 1}{4\pi} e^{ix(q+k)/2 + i(q-k)/2} \frac{E_-(k) - E_-(q)}{\sin\frac{k-q}{2}}, \qquad \Gamma = \int\limits_{-\pi}^{\pi} \frac{dk}{2\pi} e^{-ixk}(1 - e^{-2\pi i \nu(k)}), \tag{57}$$

$$\delta V_-(k,q) = \frac{e^{2\pi i \nu(k)} - 1}{2\pi}(E_-(q) - i\Gamma/2)(E_-(k) + i\Gamma/2)e^{ix(q+k)/2}, \tag{58}$$

$$E_-(q) = \int\limits_{-\pi-i0}^{\pi-i0} \frac{dk}{4\pi} e^{-ixk}(e^{-2\pi i \nu(k)} - 1)\cot\frac{k-q}{2} + ie^{-ixq}. \tag{59}$$

For large $x > 0$, we notice that $\Gamma$ is exponentially suppressed and $E_-(q) \approx ie^{-ixq}$, so $\tau_-(x)$ transforms into a generalized sine kernel Fredholm determinant Eq. (18) up to terms exponentially small in $x$. The corresponding asymptotics can be obtained in a way similar to $\delta < 0$, however instead of summation over "holes" $q_a$ we will have summation over extra particles $k_a$. We demonstrate how it works for $\delta = 2$. In this case the set $\mathbf{q}$ consists of $L$ elements and the set $\mathbf{k}$ of $L+1$ elements, which we parameterize by the omission one of the $L+2$ momenta from the all possible solutions of Eq. (6). Namely,

$$\mathbf{k}^{(a)} = \{k_1, \dots, k_{a-1}, k_{a+1}, \dots, k_{L+2}\}, \qquad a = 1, 2, \dots, L+2. \tag{60}$$

The relative momentum of this state in the thermodynamic limit reads as

$$\Delta P = \sum_{k \in \mathbf{k}^{(a)}} k - \sum_{i=1}^{L} q_i = 2\pi - k_a - \int\limits_{-\pi}^{\pi} \nu(q)dq. \tag{61}$$

The corresponding overlaps are given in Appendix C.2. Taking the thermodynamic limit we obtain the following presentation suited for the asymptotic analysis when $x \to +\infty$

$$\tau_-(x) = \mathcal{A}_- \int\limits_{-\pi}^{\pi} \frac{dk}{2\pi}(e^{-2\pi i \nu(k)} - 1)\exp\left(ik(x+2) + \int\limits_{-\pi}^{\pi}\left(\nu(q) - \frac{q}{\pi}\right)\cot\frac{q-k-i0}{2}dq\right), \tag{62}$$

$$\mathcal{A}_- = \exp\left[-\frac{1}{2}\int\limits_{-\pi}^{\pi} dq \int\limits_{-\pi}^{\pi} dk\left(\frac{\nu(q) - \nu(k) - (q-k)/\pi}{2\sin\frac{q-k}{2}}\right)^2\right]. \tag{63}$$

For $\delta > 2$ one can obtain similar determinant representation as for $\tau_-(x)$ in Eq. (46).

Even though we have constructed $\tau_-(x)$ to address positive indices $\delta > 1$ it is possible to describe $\delta < 1$ by the previous choice of $g(k)$ Eq. (15) and considering $x < 0$.

## 4 Quantum XY spin chain and its correlation functions

In this section we consider an application of the general results obtained in the previous sections to the derivation of large distance asymptotics of thermal spin-spin correlation functions

of the quantum XY spin chain. The quantum XY spin chain in a transverse field is defined by the Hamiltonian [63, 70]

$$\mathbf{H}_{\text{XY}} = -\frac{1}{2} \sum_{m=1}^{L} \left( \frac{1+\gamma}{2} \sigma_m^x \sigma_{m+1}^x + \frac{1-\gamma}{2} \sigma_m^y \sigma_{m+1}^y + h \sigma_m^z \right), \tag{64}$$

where a periodic boundary condition for the spin operators is assumed $\sigma_{L+1}^\alpha = \sigma_1^\alpha$, $\gamma$ is an anisotropy parameter, and $h$ is the strength of the magnetic field. The Hamiltonian $\mathbf{H}_{\text{XY}}$ of XY model can be considered as an anisotropic deformation of the Hamiltonian $\mathbf{H}_{\text{XX}}$ of XX model, corresponding to $\gamma = 0$. The Hamiltonian $\mathbf{H}_{\text{XX}}$ can be diagonalized in two steps: Jordan–Wigner transformation to fermionic operators and a Fourier transform to momenta representation. To diagonalize the Hamiltonian $\mathbf{H}_{\text{XY}}$ an additional Bogoliubov transformation is needed [50, 63, 70] specified by the angle $\theta(p)$:

$$e^{i\theta(p)} = \frac{h - \cos(p) - i\gamma \sin(p)}{\mathcal{E}(p)}, \qquad \mathcal{E}(p) = \sqrt{(h - \cos(p))^2 + \gamma^2 \sin^2(p)}. \tag{65}$$

Here $\mathcal{E}(p)$ stands for the spectrum of the effective Dirac fermions $A_p$, and the Hamiltonian $\mathbf{H}_{\text{XY}}$ reduces to the free-fermionic one, namely

$$\mathbf{H}_{\text{XY}} = \sum_p \mathcal{E}(p) \left( A_p^+ A_p - 1/2 \right). \tag{66}$$

We skip the details of the fermions boundary conditions as they are not important in the thermodynamic limit. We focus on the following spin-spin correlation function at finite temperature

$$G(m) \equiv \left\langle \sigma_{m+1}^x \sigma_1^x \right\rangle_T = \frac{\text{Tr}\left( \sigma_{m+1}^x \sigma_1^x e^{-\beta \mathbf{H}_{\text{XY}}} \right)}{\text{Tr}\left( e^{-\beta \mathbf{H}_{\text{XY}}} \right)}. \tag{67}$$

It is the most interesting two point correlation function as the others are either trivial in the thermodynamic limit: $\langle \sigma_{m+1}^x \sigma_1^y \rangle_T = 0$, can be expressed in terms of elementary functions as $\langle \sigma_{m+1}^z \sigma_1^z \rangle_T$ (see Ref. [60]), or related to $G(m)$ after the change $\gamma \to -\gamma$ as $\langle \sigma_{m+1}^y \sigma_1^y \rangle_T$. We follow Ref. [50] to present $G(m)$ in the thermodynamic limit as Fredholm determinants $(m > 0)$:

$$G(m) = \det(1 + \hat{W} + \widehat{\delta W}) - \det(1 + \hat{W}), \tag{68}$$

where the operators $\hat{W}$, $\widehat{\delta W}$ are integral operators on $L^2([-\pi, \pi])$ with the kernels given by

$$W(p, q) = -\frac{1}{\pi} e^{\frac{i(p-q)}{2}} \frac{\sin \frac{m(p-q)}{2}}{\sin \frac{p-q}{2}} \omega_F(q), \qquad \delta W(p, q) = \frac{1}{\pi} \exp \frac{-im(p+q)}{2} \omega_F(q), \tag{69}$$

$$\omega_F(q) = \frac{1}{2} \left( 1 - e^{i\theta(q)} \tanh \frac{\beta \mathcal{E}(q)}{2} \right). \tag{70}$$

In this form we immediately observe that $G(m)$ can be identified with $\tau_S(m)$ defined in Eq. (18) with the appropriate choice of $\nu(q)$, which can be read off from the prefactor in front of the sine-kernel

$$e^{2\pi i \nu(k)} = 1 - 2\omega_F(k) = e^{i\theta(k)} \tanh \frac{\beta \mathcal{E}(k)}{2}. \tag{71}$$

This way, to find the large $m$ asymptotics we approximate $G(m)$ by $\tau(m)$ from Eq. (11) and use results for form factor series obtained in the previous sections. The analysis depends on the winding number $\delta = \nu(\pi) - \nu(-\pi)$, which can be read off from the following form of the phase shift

$$\nu(k) = \frac{\theta(k)}{2\pi} + \frac{1}{2\pi i} \log \tanh \frac{\beta \mathcal{E}(k)}{2}. \tag{72}$$

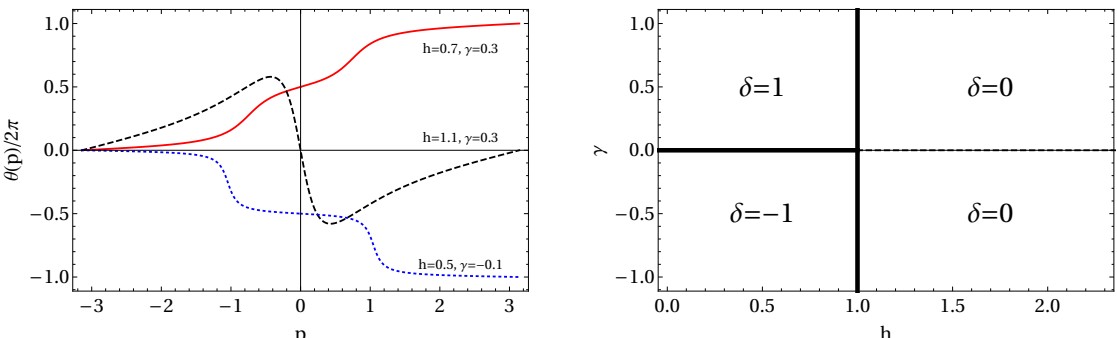

Figure 1: (left panel): the dependence of Bogoliubov angles on momentum for three different points in $h-\gamma$-plane: $h = 0.7$, $\gamma = 0.3$ ($\delta = 1$) – red solid, $h = 1.1$, $\gamma = 0.3$ ($\delta = 0$) – black dashed, $h = 0.5$, $\gamma = -0.1$ ($\delta = 0$) – blue dotted; (right panel): three regions in $h-\gamma$-plane corresponding to $\delta = \pm 1$ (ferromagnetic phase with $\gamma \gtrless 0$) and $\delta = 0$ (paramagnetic phase ).

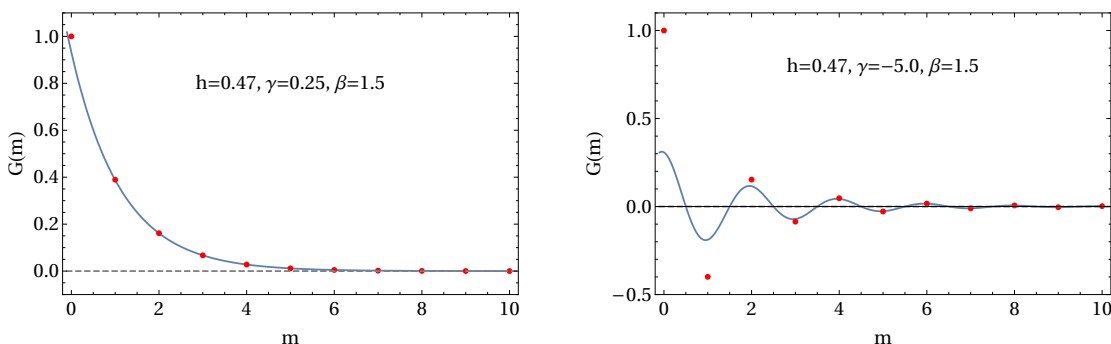

Figure 2: The exact values of the correlation function $G(m)$ (red dots) and its large distance asymptotics (blue solid curves). The left panel corresponds to $h = 0.47$, $\gamma = 0.25$, $\beta = 1.5$, and $\delta = +1$. The right panel corresponds to $h = 0.47$, $\gamma = -5.0$, $\beta = 1.5$, and $\delta = -1$.

The winding number is governed by the Bogoliubov angle $\theta(\pi) - \theta(-\pi) = 2\pi\delta$. The possible values of $\delta$ are $\delta = 0, \pm 1$ depending on the anisotropy parameter $\gamma$ and the magnetic field $h$ (see Fig. 1 for the typical behaviour of the Bogoliubov angle and the phase diagram).

Notice also that Eq. (72) implies that the integral entering the asymptotic formulas can be presented as

$$\int_{-\pi}^{\pi} dq\, \nu(q) = \pi\delta + \frac{1}{2\pi i} \int_{-\pi}^{\pi} dq\, \log\tanh\frac{\beta\mathcal{E}(q)}{2}. \tag{73}$$

In Fig. 2 we plot exact values for the correlation function $G(m)$ (red dots) and compare them with the asymptotic formulas written explicitly below (blue curves). We see that large $m$ asymptotics gives reasonable approximation even for $m \sim 1$. In fact, to get any visual discrepancy we had to consider large negative anisotropies in the ferromagnetic phase ($\delta = -1$ and $\gamma < -1$). It turns out that in this case the asymptotic formulas for non-integer $m$ acquire nonzero imaginary part, which is discarded in the plot. For integer points the imaginary part is equal to zero. Below we analyze each case separately and present analytical formulas for the asymptotics. These expressions turn out to be in accordance with the results of Ref. [60] but have a more compact form.

The results of Sec. 5 allow us to present the difference of the determinants in Eq. (68) as

a single determinant, namely

$$G(m) = \det(1 + \hat{W}_1), \tag{74}$$

where $\hat{W}_1$ is an integral operator on $L^2([-\pi, \pi])$ with the generalized sine-kernel given by

$$W_1(p, q) = \frac{e^{2\pi i \nu_1(p)} - 1}{2\pi} \frac{\sin \frac{m(p-q)}{2}}{\sin \frac{p-q}{2}} = -\frac{e^{i(\theta(p)-p)} \tanh \frac{\beta \mathcal{E}(p)}{2} + 1}{2\pi} \frac{\sin \frac{m(p-q)}{2}}{\sin \frac{p-q}{2}}, \tag{75}$$

$$\nu_1(p) = \nu(p) - \frac{p + \pi}{2\pi}. \tag{76}$$

This result is a particular case of the relation (107) with (104).

## 4.1 Paramagnetic phase $h > 1$ ($\delta = 0$)

We start our consideration with relatively large magnetic field $h > 1$. For zero temperature such values of $h$ correspond to the paramangetic phase, while for finite temperature the corresponding $\nu(q)$ has zero winding number $\delta = 0$. This way, we use formula (33) to find asymptotic behavior of the correlation function $G(m)$ at large $m$, namely

$$G(m) = \tau_S(m) \approx -T_0(m) z_1^{-m-1} \mathfrak{S}(z_1) \operatorname{res}_{z=z_1} e^{-2\pi i \nu(k)}, \qquad z = e^{ik}. \tag{77}$$

where $T_0(m)$ and $\mathfrak{S}(z)$ are given by (28) and (31), respectively. The point $z_1$ is the position of the pole of $e^{-2\pi i \nu(k)}$ outside unit circle with minimal absolute value. To find $z_1$ we factorize

$$Q(z) = \mathcal{E}^2(k) = (h - \cos k)^2 + \gamma^2 \sin^2 k = \frac{1 - \gamma^2}{4z^2}(z - x_-)(z - x_+)(z - y_-)(z - y_+), \tag{78}$$

$$x^\pm = \frac{h - \sqrt{h^2 + \gamma^2 - 1}}{1 \pm \gamma}, \qquad y^\pm = \frac{h + \sqrt{h^2 + \gamma^2 - 1}}{1 \pm \gamma}, \qquad (x^\pm)^{-1} = y^\mp. \tag{79}$$

The exponent of the angle $\theta(k)$ of Bogoliubov transformation can also be presented in a factorized form, which leads to

$$e^{-2\pi i \nu(k)} = e^{-i\theta(k)} \coth \frac{\beta \mathcal{E}(k)}{2} = -\frac{2z}{1 + \gamma} \frac{\sqrt{Q(z)} \coth \frac{\beta \sqrt{Q(z)}}{2}}{(z - x_+)(z - y_+)}. \tag{80}$$

It useful to present $\sqrt{Q(z)} \coth \frac{\beta}{2} \sqrt{Q(z)}$ as an infinite product

$$\sqrt{Q(z)} \coth \frac{\beta}{2} \sqrt{Q(z)} = \frac{2}{\beta} \frac{\prod_{n=1}^\infty \left(1 + \frac{\beta^2 Q(z)}{(2n-1)^2 \pi^2}\right)}{\prod_{n=1}^\infty \left(1 + \frac{\beta^2 Q(z)}{(2n)^2 \pi^2}\right)}. \tag{81}$$

Notice that in such a form the branch cut singularities disappear manifestly. Moreover, the analysis of the poles of $e^{-2\pi i \nu(k)}$ is now a straightforward task, from which we conclude that the smallest (by the absolute value) pole outside the unit circle is $z_1 = y_+$ for all non-zero temperatures. Therefore using Eq. (80) and Eq. (81) we obtain

$$\operatorname{res}_{z=y_+} e^{-2\pi i \nu(k)} = -\frac{2}{\beta} \frac{y_+}{\sqrt{h^2 + \gamma^2 - 1}}. \tag{82}$$

Finally, taking into account (73) for $\delta = 0$, the asymptotics reads

$$G(m) \approx \mathcal{A} e^{-m/\xi}, \tag{83}$$

where

$$\xi^{-1} = \log y_+ - \frac{1}{2\pi} \int\limits_{-\pi}^{\pi} dq \, \log \tanh \frac{\beta \mathcal{E}(q)}{2}, \qquad y_+ = \frac{h + \sqrt{h^2 + \gamma^2 - 1}}{1 + \gamma}, \tag{84}$$

$$\mathcal{A} = \frac{2}{\beta \sqrt{h^2 + \gamma^2 - 1}} \exp\left( -\frac{1}{2} \int\limits_{-\pi}^{\pi} dq \int\limits_{-\pi}^{\pi} dp \left[ \frac{\nu(q) - \nu(p)}{2 \sin \frac{q-p}{2}} \right]^2 + i \int\limits_{-\pi}^{\pi} dq \, \nu(q) \frac{y_+ + e^{iq}}{y_+ - e^{iq}} \right). \tag{85}$$

The sign of the magnetic field $h$ is irrelevant since it can be flipped by the conjugation of the Hamiltonian with $\sigma^x$ acting in each site. Therefore below we consider $0 < h < 1$.

## 4.2  Ferromagnetic phase $h < 1$, $\gamma > 0$ ($\delta = 1$)

In the ferromagnetic phase $h < 1$ with positive anisotropy $\gamma > 0$ we use the asymptotics (22) and the integral (73) for $\delta = 1$ to obtain

$$G(m) \approx \mathcal{A} e^{-m/\xi}, \tag{86}$$

with

$$\xi^{-1} = -\frac{1}{2\pi} \int\limits_{-\pi}^{\pi} dq \, \log \tanh \frac{\beta \mathcal{E}(q)}{2}, \tag{87}$$

$$\mathcal{A} = \exp\left( -\frac{1}{2} \int\limits_{-\pi}^{\pi} dq \int\limits_{-\pi}^{\pi} dp \left[ \frac{\nu(q) - \nu(p) - (q-p)/2\pi}{2 \sin \frac{q-p}{2}} \right]^2 \right). \tag{88}$$

For particular values of the parameters we plot exact correlation function $G(m)$ and its asymptotics (86) in the left panel of Fig. 2.

Note that even though formulas for the correlation length in different parameter regions Eq. (84) and Eq. (87) look different, the transition $h < 1$ and $h > 1$ is analytic in $h$. The same is true for prefactors $\mathcal{A}$ given by Eq. (85) and Eq. (88) (see the corresponding plots in Fig. 3). This reflects the fact that at finite temperature in one dimensional systems with short-range interactions phase transitions are absent and the physical observables are smooth functions of system parameters. This observation was used in Ref. [71] to obtain correct expressions for the correlation length and prefactor for the Ising model in the scaling limit.

## 4.3  Ferromagnetic phase $h < 1$, $\gamma < 0$ ($\delta = -1$)

In this region of parameters, the correlation function $G(m)$ is given by Eq. (46) for $n = 2$. We will need the large $m$ asymptotics of $Y_{-1}(m)$, which for $h^2 + \gamma^2 \neq 1$ is given by

$$Y_{-1}(m) \approx A_1 e^{-\varkappa_1 m} + A_2 e^{-\varkappa_2 m}, \tag{89}$$

where, as seen from Eq. (79),

$$\varkappa_1 = \log x_+, \qquad \varkappa_2 = \log y_+, \tag{90}$$

$$A_1 = \frac{2}{\beta} \frac{1}{\sqrt{h^2 + \gamma^2 - 1}} \frac{1}{x_+} \mathfrak{S}_{-1}(x_+), \qquad A_2 = -\frac{2}{\beta} \frac{1}{\sqrt{h^2 + \gamma^2 - 1}} \frac{1}{y_+} \mathfrak{S}_{-1}(y_+), \tag{91}$$

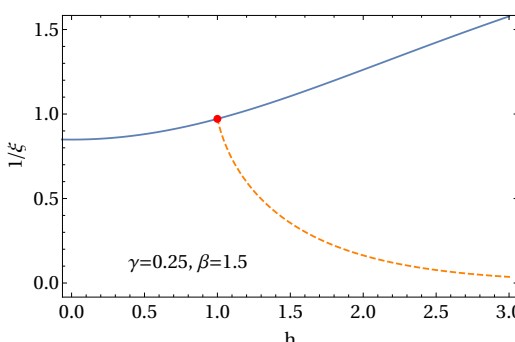
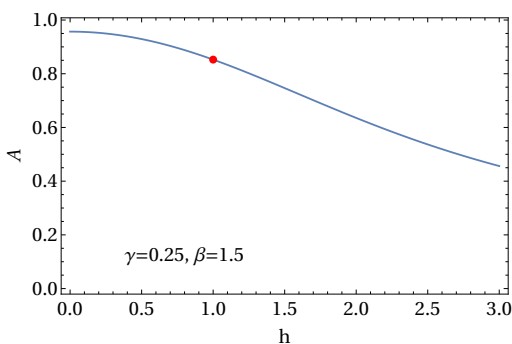

Figure 3: The inverse correlation length (left panel) and the prefactor (right panel) for different values of magnetic field $h$. Blue solid curves correspond to Eq. (87) [Eq. (88)] for $h < 1$ and Eq. (84) [Eq. (85)] for $h > 1$, for the left [right] panels, respectively. The orange line shows formal use of Eq. (87) for the region $h > 1$.

$$\mathfrak{S}_{-1}(z) = \exp\left( i \int_{-\pi}^{\pi} dq \left( \nu(q) + \frac{\pi + q}{2\pi} \right) \frac{z + e^{iq}}{z - e^{iq}} \right). \tag{92}$$

Therefore Eq. (50) becomes

$$\left| \begin{matrix} Y_{-1}(m) & Y_{-1}(m+1) \\ Y_{-1}(m-1) & Y_{-1}(m) \end{matrix} \right| \approx \frac{16}{\beta^2 (1-\gamma)^2} \mathfrak{S}_{-1}(x_+)\mathfrak{S}_{-1}(y_+) e^{-m(\log x_+ + \log y_+)}. \tag{93}$$

Finally, the large distance asymptotic for $G(m)$ following from (46) is

$$G(m) \approx \mathcal{A} e^{-m/\xi}, \tag{94}$$

where

$$\xi^{-1} = \log x_+ + \log y_+ - \frac{1}{2\pi} \int_{-\pi}^{\pi} dq \, \log \tanh \frac{\beta \mathcal{E}(q)}{2}, \tag{95}$$

$$\mathcal{A} = \frac{16}{\beta^2 (1-\gamma)^2} \mathfrak{S}_{-1}(x_+)\mathfrak{S}_{-1}(y_+) \exp\left( -\frac{1}{2} \int_{-\pi}^{\pi} dq \int_{-\pi}^{\pi} dp \left[ \frac{\nu(q) - \nu(p) + (q-p)/(2\pi)}{2 \sin \frac{q-p}{2}} \right]^2 \right). \tag{96}$$

In the case when $h^2 + \gamma^2 = 1$ we have $x_+ = y_+$ and the derivation is changed slightly (in particular, $Y_{-1}(m) \approx (B + Cm)e^{-m \log x_+}$) however the final formula for the asymptotic of $G(m)$ is the same. Notice that for non-integer values of $m$ the right hand side of Eq. (94) becomes a complex valued function. We plot the typical behaviour of $G(m)$ and the real part of its asymptotics in (94) in the right panel of Fig. 2.

## 5 Relation to Toeplitz determinants

The traditional approach to the correlation functions in the XY spin chain is in presenting them via Toeplitz determinants [60, 63]. Asymptotic analysis of these structures can be performed by means of the Szegő theorem [72, 73] and its generalization[3] by Hartwig and Fisher [61]. Let

---

[3]Here we focus only on the smooth symbols with the only "singularity" given by the non-trivial winding number

us comment on how similar structures can appear within our effective form factors approach. In addition to tau functions (5) and (54) that contained different number of "particles" in bra- and ket- states, we define

$$\tau_0(x) = \sum_{\mathbf{q}} |\langle \mathbf{p} | \mathbf{q} \rangle|^2 e^{-ix\left(\sum_{i=1}^{N} p_i - \sum_{i=1}^{N} q_i\right)}, \tag{97}$$

where the quasi momenta $q$ are solutions of $e^{iqL} = 1$, while $p$ are solutions of the following equation

$$e^{ipL} = e^{-2\pi i \omega(p)}. \tag{98}$$

Here for convenience we have chosen a different notation for the phase shift. We focus on the case of non-positive winding numbers for this function i.e. $\omega(\pi) - \omega(-\pi) \leq 0$. The corresponding form factors read

$$|\langle \mathbf{p} | \mathbf{q} \rangle|^2 = \frac{\prod_{i=1}^{N} e^{g_\omega(p_i) - g_\omega(q_i)}}{\prod_{i=1}^{N}(1 + \frac{2\pi}{L}\omega'(p_i))} \left(\prod_{i=1}^{N} \frac{\sin \pi \omega(p_i)}{L}\right)^2 \frac{\prod_{i>j}^{N} \sin^2 \frac{p_i - p_j}{2} \prod_{i>j}^{N} \sin^2 \frac{q_j - q_i}{2}}{\prod_{i,j=1}^{N} \sin^2 \frac{p_i - q_j}{2}}. \tag{99}$$

The summation in Eq. (97) can be performed using techniques developed in Appendix A, which together with the identification

$$e^{-g_\omega(p)} = e^{-2\pi i \omega(p)} - 1 \tag{100}$$

leads to the Fredholm determinant expression of $\tau_0$

$$\tau_0(x) = \det(1 + \hat{V}_\omega), \qquad \hat{V}_\omega = \hat{S}_\omega + \hat{R}_\omega, \tag{101}$$

where

$$S_\omega(p,q) = \frac{e^{2\pi i \omega(p)} - 1}{2\pi} \frac{\sin \frac{x(p-q)}{2}}{\sin \frac{p-q}{2}}, \qquad R_\omega(p,q) = \frac{e^{2\pi i \omega(p)} - 1}{4\pi} e^{-i(p+q)x/2} \frac{r_\omega(p) - r_\omega(q)}{\sin \frac{p-q}{2}}, \tag{102}$$

$$r_\omega(k) = \int_{-\pi}^{\pi} \frac{dq}{4\pi} (e^{-2\pi i \omega(q)} - 1) e^{iqx} \cot \frac{q + i0 - k}{2}. \tag{103}$$

Notice that definitions of the kernels of $\hat{V}$ and $\hat{S}_\nu$ differ from their analogues introduced in Sec. 2 by the conjugation with diagonal matrices, which does not change the value of the determinant. Comparing overlaps (9) and (99) (see Appendix C.5) we conclude that imposing the following relation between $\nu(q)$ and $\omega(q)$

$$\omega(q) = \nu(q) - \frac{q + \pi}{2\pi} \equiv \nu_1(q), \tag{104}$$

we obtain exact equality for the tau functions, namely

$$\det\left(1 + \hat{V}_\nu + \delta \hat{V}_\nu\right) - \det\left(1 + \hat{V}_\nu\right) = \det\left(1 + \hat{V}_{\nu_1}\right). \tag{105}$$

Here the finite rank contribution is modified due to the conjugation with the diagonal matrices

$$\delta V_\nu(p,q) = -\frac{e^{2\pi i \nu(p)} - 1}{2\pi} e^{-i(x+1)p/2} e^{-i(x-1)q/2}. \tag{106}$$

Similar relations can be obtained between $\tau_-(x)$ and $\tau_0(x)$ for $\delta > 1$. For large positive $x$, functions $r_\omega(x)$ are exponentially small, so Eq. (107) holds for the generalized sine-kernels $\hat{S}_\nu$.

$$\det\left(1 + \hat{S}_\nu + \delta\hat{V}_\nu\right) - \det\left(1 + \hat{S}_\nu\right) = \det\left(1 + \hat{S}_{\nu_1}\right). \tag{107}$$

In fact we can easily demonstrate that this relation is true for any positive integer $x$. To do so we will clarify the relation between Fredholm and Toeplitz determinants (cf. [62,74]). It is convenient to deform slightly the kernel by the set of functions $a_0(p), a_1(p), ..., a_{x-1}(p)$

$$S_\nu^a(p,q) = \frac{e^{2\pi i \nu(p)} - 1}{2\pi} \sum_{n=0}^{x-1} a_n(p) e^{in(q-p)}. \tag{108}$$

For $a_i(q) = 1$ one can easily see that we recover the kernel of $\hat{S}_\nu$ up to conjugation with diagonal matrices, which does not affect the value of the determinant

$$\det\left(1 + \hat{S}_\nu\right) = \det\left(1 + \widehat{S^a}\right)\Big|_{a_0=a_1=...a_{x-1}=1}. \tag{109}$$

Furthermore, we can treat $\widehat{S^a}$ as a product of two rectangular matrices

$$\widehat{S^a} = \mathcal{A}\mathcal{B}, \qquad \mathcal{A}_{qn} = e^{iqn}, \qquad \mathcal{B}_{np} = \frac{e^{2\pi i \nu(p)} - 1}{2\pi} a_n(p) e^{-inp}. \tag{110}$$

Then using the fact that $\det\left(1 + \mathcal{A}\mathcal{B}\right) = \det\left(1 + \mathcal{B}\mathcal{A}\right)$ we obtain a relation between the Fredholm determinant and determinant of matrix $x$ by $x$, namely

$$\det\left(1 + \widehat{S^a}\right) = \det_{0 \le n,m \le x-1}\left(\delta_{nm} + T_{nm}\right), \qquad T_{nm} = \int_{-\pi}^{\pi} \frac{dq}{2\pi} a_n(q)(e^{2\pi i \nu(q)} - 1)e^{-i(n-m)q}. \tag{111}$$

For $a_n = 1$ the matrix $T_{nm}$ transforms into the Toeplitz one, namely

$$\det\left(1 + \widehat{S^a}\right) = \det_{0 \le n,m \le x-1} c_{n-m}, \qquad c_k = \int_{-\pi}^{\pi} \frac{dq}{2\pi} e^{2\pi i \nu(q)} e^{-ikq}. \tag{112}$$

In order to account for the finite rank we notice that because rank-one contributions are at most linear in the determinant expansion, we can present

$$\det\left(1 + \hat{S}_\nu + \delta\hat{V}_\nu\right) - \det\left(1 + \hat{S}_\nu\right) = \frac{\partial}{\partial\alpha}\det\left(1 + \hat{S}_\nu + \alpha\delta\hat{V}_\nu\right)\Big|_{\alpha=0}. \tag{113}$$

To account for the finite $\alpha$ one must choose $a_0(q) = 1 - \alpha e^{-ixq}$ and $a_n(q) = 1$ for $n \ge 1$, therefore

$$\det\left(1 + \hat{S}_\nu + \alpha\delta\hat{V}_\nu\right) = \det\left(1 + \widehat{S^a}\right) = \det\begin{pmatrix} c_0 - \alpha c_x & c_{-1} - \alpha c_{x-1} & \cdots & c_{-x+1} - \alpha c_1 \\ c_1 & c_0 & \cdots & c_{-x+2} \\ \cdot & \cdot & & \cdot \\ \cdot & \cdot & \ddots & \cdot \\ \cdot & \cdot & & \cdot \\ c_{x-1} & c_{x-2} & \cdots & c_0 \end{pmatrix}. \tag{114}$$

Since we are looking only to the terms linear in $\alpha$, we can leave only terms that are proportional to $\alpha$ in the first row. Moreover we can replace this row with the last one. This way we obtain

$$\frac{\partial}{\partial \alpha} \det\left(1 + \hat{S}_\nu + \alpha\delta\hat{V}_\nu\right)\Big|_{\alpha=0} = (-1)^x \det\begin{pmatrix} c_1 & c_0 & \dots & c_{-x+2} \\ c_2 & c_1 & \dots & c_{-x+3} \\ . & . & & . \\ . & . & \ddots & . \\ . & . & & . \\ c_x & c_{x-1} & \dots & c_1 \end{pmatrix} = \det_{0 \leq n,m \leq x-1} \tilde{c}_{n-m}, \quad (115)$$

where

$$\tilde{c}_k = -\int_{-\pi}^{\pi} \frac{dq}{2\pi} e^{2\pi i\, v(q)} e^{-i(k+1)q} = \int_{-\pi}^{\pi} \frac{dq}{2\pi} e^{2\pi i\, v_1(q)} e^{-ikq}. \quad (116)$$

Here we see the shift $v(q) \to v_1(q)$ as predicted from the finite size scaling of the form factors in Eq. (104). This shift together with Eq. (112) completes the proof of Eq. (107).

Let us also comment on how results of Sec. 3.3 reproduce Hartwig and Fisher asymptotic behaviour (Theorem 4 in Ref. [61]). As $v_\delta(q)$ has zero winding number we can expand it as

$$v_\delta(q) = \frac{-1}{2\pi i} \sum_{n=-\infty}^{\infty} k_n e^{iqn}. \quad (117)$$

Then the integral in the exponential in Eq. (48) can be evaluated as

$$-\int_{-\pi}^{\pi} dp\, v_\delta(p) \cot\frac{q-p+i0}{2} = \int_{-\pi}^{\pi} \frac{dp}{2\pi} \frac{e^{i(q+i0)} + e^{ip}}{e^{i(q+i0)} - e^{ip}} \sum_{n=-\infty}^{\infty} k_n e^{ipn} = -k_0 - 2\sum_{n=1}^{\infty} e^{iqn} k_n. \quad (118)$$

In this derivation we used that $|e^{i(q+i0)}| < 1$ and expanded the denominator as a geometric series. Substituting this result back into Eq. (48) we immediately see that $Y_\delta(x) = l_x$, where the Fourier modes $l_m$ are defined through the relation

$$\exp\left(\sum_{n=1}^{\infty}\left(k_{-n} e^{-iqn} - k_n e^{iqn}\right)\right) = \sum_{m=-\infty}^{\infty} l_m e^{imq}. \quad (119)$$

Finally, expressing double integral in the asymptotic expression Eq. (46)

$$-\frac{1}{2}\int_{-\pi}^{\pi} dq \int_{-\pi}^{\pi} dp \left[\frac{v_\delta(q) - v_\delta(p)}{2\sin\frac{q-p}{2}}\right]^2 = \sum_{n=1}^{\infty} n k_n k_{-n}, \quad (120)$$

we obtain the statement of Theorem 4 in Ref. [61].

# 6  Summary and Outlook

In this work we have introduced the form factors (overlaps) to simulate the static correlation functions for the states with finite entropy. The state was determined by the phase shift function $v(q)$. For the traditional approaches dealing with the finite entropy states is notoriously difficult but for our approach it is rather advantageous situation, since almost all available quantum numbers are occupied which tremendously simplifies the computation of form factor series. This allows us, in particular, to re-derive known asymptotics for the static two point

correlators in the XY spin chain and present them in a more compact form. We hope that the simplicity of this approach will make it possible to obtain the full asymptotic expansion at large distances.

Apart from the thermal state we can apply our approach to the states resulting from the long time evolution after a quench [75–80], to models of 1D anyons [81–86], or mobile impurity models [64, 87, 88]. This can be done by the appropriate modification of the phase shift function. We will discuss it elsewhere.

It is interesting to note that $\nu(q)$ is apparently connected with the auxiliary functions that appears in the Quantum Transfer Matrix (QTM) approach and specifies the Bethe roots for QTM [38–40, 89]. It would be interesting to completely clarify connection between these two approaches.

The correlation functions at zero temperature (entropy) can be formally accounted by the jump discontinuities in $\nu(q)$, which can also be treated by the form factor summation developed for the critical models [11, 13]. In this case the role of the lattice is not essential and the exponential asymptotic behaviour is expected to be replaced by a power-law, which can be obtained from the proper modification of the generalized sine-kernels (see section 9 in Ref. [90]). To address dynamical correlation functions we must modify appropriately the form factors and the spectral factor $e^{-i\sum k_i x} \to e^{-i\sum(k_i x - \epsilon(k_i)t)}$. The detailed constructions and extraction of the asymptotic behavior is another intriguing direction for future research. However we can already anticipate that for the space-like region, i.e. when the saddle point of the expression $kx - \epsilon(k)t$ is outside the Brillouin zone, the asymptotic analysis remains largely unchanged, which can be immediately seen in the asymptotics of Ref. [59]. For the time-like region the main problem will be that a suitable $\nu(q)$ might have a jump discontinuity which leads to additional power-law behavior (c.f. Ref. [91]). Finally, there will be extra $1/\sqrt{t}$ terms connected to the saddle point contributions indicated by the non-linear Luttinger theory [14–16].

## Acknowledgements

We are grateful to Nikita Slavnov and Frank Göhmann for useful discussions. We thank Oleg Lychkovskiy and Daniel Chernowitz for careful reading of the manuscript and numerous useful remarks and suggestions. The authors acknowledge the support by the National Research Foundation of Ukraine grant 2020.02/0296. O. G. acknowledges the support from the European Research Council under ERC Advanced grant 743032 DYNAMINT.

## A  Summation of form factors and determinant formula

In this appendix, we derive formula (11) presenting tau function in the thermodynamic limit as a difference of two Fredholm determinants.

We consider solutions in the large $L$ limit and choose **k** to fill a Fermi Sea, namely

$$k_i = \frac{2\pi}{L}\left(-\frac{N}{2} + i - 1 - \nu_i\right), \qquad i = 1, \dots, N+1, \tag{121}$$

where $\nu_i \equiv \nu(k_i)$. For simplicity, we choose $N$ to be even.

First, we identically rewrite the overlap as (note, $\det D = \det \tilde{D}$)

$$|\langle \mathbf{k}|\mathbf{q}\rangle|^2 = -4L \prod_{i=1}^{N+1} \Omega_i \left(\prod_{i=1}^{N+1} \frac{e^{g(k_i)/2}\sin\pi\nu_i}{L}\right)^2 \prod_{i=1}^{N} e^{-g(q_i)} \det D \det \tilde{D} \tag{122}$$

$$D = \begin{pmatrix} \cot \frac{k_1-q_1}{2} - i & \dots & \cot \frac{k_{N+1}-q_1}{2} - i \\ \vdots & \ddots & \vdots \\ \cot \frac{k_1-q_N}{2} - i & \dots & \cot \frac{k_{N+1}-q_N}{2} - i \\ 1 & \dots & 1 \end{pmatrix}, \quad \tilde{D} = \begin{pmatrix} \cot \frac{k_1-q_1}{2} + i & \dots & \cot \frac{k_{N+1}-q_1}{2} + i \\ \vdots & \ddots & \vdots \\ \cot \frac{k_1-q_N}{2} + i & \dots & \cot \frac{k_{N+1}-q_N}{2} + i \\ 1 & \dots & 1 \end{pmatrix},$$

$$\tag{123}$$

$$\Omega_i = \frac{1}{1 + \frac{2\pi \nu'(k_i)}{L}}. \tag{124}$$

Then using standard linear algebra manipulations we rewrite the static tau function as

$$\tau(x) = \det(\mathcal{A} + \delta\mathcal{A}) - \det \mathcal{A} \tag{125}$$

with

$$\delta \mathcal{A}_{ij} = -\frac{4\Omega_i}{L} \sin^2(\pi \nu_i) e^{g(k_i)} e^{-i(k_i+k_j)x/2}, \tag{126}$$

$$\mathcal{A}_{ij} = \Omega_i \frac{\sin^2(\pi \nu_i)}{L^2} e^{g(k_i)} e^{-i(k_i+k_j)x/2} \sum_q e^{iqx-g(q)} \left(\cot \frac{q-k_i}{2} - i\right)\left(\cot \frac{q-k_j}{2} + i\right), \tag{127}$$

where summation over $q$ is happening over the whole Brillouin zone

$$q \in \left\{\frac{2\pi}{L}\left(-\frac{L-1}{2} + j - 1\right), \quad j = 1, \dots, L\right\}. \tag{128}$$

For $i \neq j$ we present

$$\mathcal{A}_{ij} = \Omega_i \frac{\sin^2 \pi \nu_i}{2L} e^{g(k_i)} e^{-i(k_i+k_j)x/2} e^{i(k_i-k_j)/2} \frac{c(k_i) - c(k_j)}{\sin \frac{k_i-k_j}{2}} \tag{129}$$

with

$$c(k_i) = \frac{2}{L} \sum_q e^{iqx-g(q)} \cot \frac{q-k_i}{2}. \tag{130}$$

This sum can be rewritten as a contour integral and evaluated at large $L$, namely, choosing contour $\gamma$ running around $q_i$ and avoiding any other singularities of the integrand we obtain

$$c(k_i) = \oint_\gamma \frac{dq}{\pi} \frac{e^{-g(q)+iqx}}{e^{iqL} - 1} \cot \frac{q-k_i}{2}. \tag{131}$$

Further, we deform the contour into the rectangle that encapsulates interval $[-\pi, \pi]$. The vertical parts of this rectangle cancel and we are left with two lines above and below the real axis along with the contribution from the pole at $q = k_i$

$$c(k_i) = \left(\int_{-\pi-i0}^{\pi-i0} - \int_{-\pi+i0}^{\pi+i0}\right) \frac{dq}{\pi} \frac{e^{-g(q)+iqx}}{e^{iqL}-1} \cot \frac{q-k_i}{2} - \frac{4ie^{-g(k_i)+ik_ix}}{e^{ik_iL}-1}. \tag{132}$$

Here we assume that the imaginary shift $i0$ is chosen to be larger then $\operatorname{Im} k_i = O(1/L)$. In this form we immediately see that, in the limit $L \to \infty$, the values of $c(k)$ at points $k_i$ are equal to the values of the $E(k_i)$ for the analytic function $E(k)$ given by

$$E(k) = \int_{-\pi+i0}^{\pi+i0} \frac{dq}{\pi} e^{-g(q)+iqx} \cot \frac{q-k}{2} - \frac{4ie^{-g(k)+ikx}}{e^{-2\pi i\nu(k)}-1}. \tag{133}$$

Using $E(k)$ we can obtain values also for some vicinity of $k_i$, which allow us to effectively "omit" solving Bethe equations (6). Performing similar computation for the diagonal components we arrive at

$$\mathcal{A}_{ii} = e^{g(k_i)-ik_ix}\frac{\Omega_i \sin(\pi\nu_i)^2}{L^2}\sum_q \frac{e^{iqx-g(q)}}{\sin^2\frac{q-k_i}{2}}. \tag{134}$$

The sum can be evaluated in the same way as in Eq. (131):

$$\mathcal{A}_{ii} = \Omega_i\left(1 + \frac{(x+ig'(k_i))(e^{2\pi i\nu(k_i)}-1)}{L}\right) + e^{g(k_i)-ik_ix}\frac{\Omega_i \sin(\pi\nu_i)^2}{2L}\int_{-\pi}^{\pi}\frac{dq}{\pi}\frac{e^{-g(q)+iqx}}{\sin^2\frac{q+i0-k_i}{2}}. \tag{135}$$

Equivalently, using definition (133), we can present

$$\mathcal{A}_{ii} = \Omega_i\left(1 + \frac{2\pi\nu'(k_i)}{L}\right) + e^{g(k_i)-ik_ix}\frac{\Omega_i \sin(\pi\nu_i)^2}{2L}2E'(k_i). \tag{136}$$

So recalling definition of (124) we obtain for generic $i$ and $j$

$$\mathcal{A}_{ij} = \delta_{ij} + \frac{\sin^2(\pi\nu_i)}{2L}e^{g(k_i)}e^{-i(k_i+k_j)x/2}e^{i(k_i-k_j)/2}\frac{E(k_i)-E(k_j)}{\sin\frac{k_i-k_j}{2}} + O(1/L^2), \tag{137}$$

where for $i = j$ the second term is understood in the L'Hopital rule sense. Similarly we obtain for the finite rank contribution

$$\delta\mathcal{A}_{ij} = -\frac{4}{L}\sin^2(\pi\nu_i)e^{g(k_i)}e^{-i(k_i+k_j)x/2} + O(1/L^2). \tag{138}$$

In this form we are at the position to take limit $L\to\infty$, and taking into account that $k_i$ is quantized in the units $2\pi/L$, arrive at the Fredholm determinants (11).

Similarly, we can perform summation for $\tau_-(x)$ defined in Eq. (54). Instead of Eq. (125) we obtain the following

$$\tau_-(x) = \det(\mathcal{A}+\delta\mathcal{A}) + (\Gamma-1)\det\mathcal{A}, \tag{139}$$

where

$$\mathcal{A}_{ij} = \frac{1}{L^2}e^{-g(q_i)+ix(q_i+q_j)/2}\sum_k\frac{e^{g(k)-ixk}\sin^2\pi\nu(k)}{1+\frac{2\pi}{L}\nu'(k)}\left(\cot\frac{k-q_i}{2}-i\right)\left(\cot\frac{k-q_j}{2}+i\right), \tag{140}$$

$$\delta A_{ij} = \frac{4}{L}F_+(q_i)F_-(q_i), \qquad \Gamma = -\frac{4}{L}\sum_k\frac{e^{g(k)-ixk}\sin^2\pi\nu(k)}{1+\frac{2\pi}{L}\nu'(k)}, \tag{141}$$

$$F_\pm(q) = e^{-\frac{g(q)}{2}+ix\frac{q}{2}}\frac{1}{L}\sum_k\frac{e^{g(k)-ixk}\sin^2\pi\nu(k)}{1+\frac{2\pi}{L}\nu'(k)}\left(\cot\frac{k-q}{2}\pm i\right). \tag{142}$$

Here $\sum_k$ means sum over all $L+\delta$ nonequivalent $(\bmod\,2\pi)$ solutions of Eq. (6), which can be presented as a contour integral

$$\frac{1}{L}\sum_k\frac{f(k)}{1+\frac{2\pi\nu'(k)}{L}} = \oint_C\frac{dk}{2\pi}\frac{f(k)}{e^{ikL+2\pi i\nu(k)}-1}, \tag{143}$$

where the contour $C$ runs around poles of the denominator only and avoids and singularities of $f(k)$. Then the derivation goes along the lines as for $\tau(x)$. Namely, for $i\neq j$ we present

$$\mathcal{A}_{ij} = \frac{1}{2L}e^{-g(q_i)+ix(q_i+q_j)/2}e^{i(q_i-q_j)/2}\frac{c(q_i)-c(q_j)}{\sin\frac{q_i-q_j}{2}}, \tag{144}$$

where now instead of Eq. (130)

$$c(q) = \frac{2}{L} \sum_k \frac{e^{g(k)-ikx} \sin^2(\pi \nu(k))}{1 + \frac{2\pi \nu'(k)}{L}} \cot \frac{k-q}{2}. \tag{145}$$

In the thermodynamic limit this function can be replaced by $E(q)$, which does not depend on the system size

$$c(q) \approx E_-(q) = \frac{1}{\pi} \int_{-\pi+i0}^{\pi+i0} dk\, e^{g(k)-ixk} \sin^2 \pi \nu(k) \cot \frac{k-q}{2} - 4i \frac{e^{g(q)-ixq} \sin^2 \pi \nu(q) e^{-2\pi i \nu(q)}}{1 - e^{-2\pi i \nu(q)}}. \tag{146}$$

For positive $x$ it is much more convenient to rewrite this function as

$$E_-(q) = \frac{1}{\pi} \int_{-\pi-i0}^{\pi-i0} dk\, e^{g(k)-ixk} \sin^2 \pi \nu(k) \cot \frac{k-q}{2} - 4i e^{g(q)-ixq} \sin^2 \pi \nu(q) \left( 1 + \frac{e^{-2\pi i \nu(q)}}{1 - e^{-2\pi i \nu(q)}} \right). \tag{147}$$

Now if we relate

$$e^{-g(q)} = e^{2\pi i \nu(q)} - 1, \tag{148}$$

this function transform into

$$E_-(q) = \int_{-\pi-i0}^{\pi-i0} \frac{dk}{4\pi} e^{-ixk} (e^{-2\pi i \nu(k)} - 1) \cot \frac{k-q}{2} + ie^{-ixq}. \tag{149}$$

For large positive $x$ the integral can be neglected. For diagonal components we obtain

$$A_{ii} = 1 + \frac{1}{L} e^{-g(q_i)+ixq_i} E'_-(q_i). \tag{150}$$

Function $\Gamma$ can be written as

$$\Gamma = \int_{-\pi}^{\pi} \frac{dk}{2\pi} e^{-ixk} (1 - e^{-2\pi i \nu(k)}). \tag{151}$$

It is also exponentially suppressed for $x \to +\infty$. The finite rank contribution is easily evaluated taking into account that

$$F_\pm(q) = \frac{e^{-g(q)/2+ixq/2}}{2} (E_-(q) \mp i\Gamma/2). \tag{152}$$

After all these transformations one readily obtains the result Eq. (56) in the thermodynamic limit.

## B  Lemmas about products

In this appendix, we study products that appear in the overlaps. In this section we assume that $\nu(q)$ is a smooth function on the segment $[-\pi, \pi]$ and assign its values in specific points as $\nu_j$, namely

$$\nu_j = \nu(q_j), \qquad q_j = \frac{2\pi}{L} \left( -\frac{L+1}{2} + j \right), \qquad \nu_- = \nu(-\pi), \qquad \nu_+ = \nu(\pi), \qquad \delta \equiv \nu_+ - \nu_-. \tag{153}$$

First we consider constant function $\nu(q) = \nu = \text{const}$.

**Lemma B.1.** *The following product formula is valid*

$$B_L(\nu) \equiv \prod_{j=1}^{L-1} \frac{\sin \frac{\pi(j-\nu)}{L}}{\sin \frac{\pi j}{L}} = \frac{\sin(\pi \nu)}{L \sin \frac{\pi \nu}{L}}. \tag{154}$$

*In the limit $L \to \infty$ this product simplifies to*

$$B_L(\nu) \approx \frac{\sin(\pi \nu)}{\pi \nu}. \tag{155}$$

*The denominator is equal to*

$$\prod_{j=1}^{L-1} \sin \frac{\pi j}{L} = \frac{L}{2^{L-1}}. \tag{156}$$

*Proof.* We can rewrite identically the left hand side as

$$B_L(\nu) = \prod_{j=1}^{L-1} \frac{\sin \frac{\pi(j-\nu)}{L}}{\sin \frac{\pi j}{L}} = e^{-i\pi\nu\frac{L-1}{L}} \prod_{j=1}^{L-1} \frac{e^{\frac{2\pi i}{L}j} - e^{\frac{2\pi i\nu}{L}}}{e^{\frac{2\pi i}{L}j} - 1}. \tag{157}$$

Taking into account that

$$\prod_{j=1}^{L-1} \left( z - e^{2\pi i j/L} \right) = \frac{z^L - 1}{z - 1}, \tag{158}$$

we obtain

$$B_L(\nu) = \frac{\sin(\pi \nu)}{L \sin \frac{\pi \nu}{L}}. \tag{159}$$

$\square$

Further, we proceed with the generic function $\nu(q)$.

**Lemma B.2.** *For an integer $0 \le A \le L-1$, the following asymptotic approximation in the limit $L \to \infty$ is valid*

$$B_{A,L}[\nu(q)] \equiv \prod_{j=1}^{A} \frac{\sin \frac{\pi(j-\nu_j)}{L}}{\sin \frac{\pi j}{L}} \approx L^{\nu_A} \Gamma \left[ \begin{array}{c} A+1-\nu_1,\ L-A \\ A+1,\ 1-\nu_1,\ L-A+\nu_A \end{array} \right] \exp \int_{-\pi}^{q_A} f(q)\, dq, \tag{160}$$

*where*

$$\Gamma \left[ \begin{array}{cccc} a_1, & a_2, & \dots & a_p \\ b_1, & b_2, & \dots & b_q \end{array} \right] = \frac{\Gamma(a_1)\Gamma(a_2)\dots\Gamma(a_p)}{\Gamma(b_1)\Gamma(b_2)\dots\Gamma(b_q)}, \tag{161}$$

$$f(q) = -\frac{\nu(q_A)}{\pi - q} + \frac{\nu_-}{\pi + q} + \frac{\nu(q)}{2} \tan \frac{q}{2}. \tag{162}$$

*Proof.* First, we introduce the modified product

$$\tilde{B}_{A,L}[\nu(q)] = \prod_{j=1}^{A} \frac{\sin \frac{\pi(j-\nu_j)}{L}}{\sin \frac{\pi j}{L}} \frac{1}{1 - \frac{\nu_j}{j}} \frac{1}{1 + \frac{\nu_j}{L-j}} = \prod_{j=1}^{A} \frac{\Gamma(1+\frac{j}{L})}{\Gamma(1+\frac{j-\nu_j}{L})} \frac{\Gamma(2-\frac{j}{L})}{\Gamma(2-\frac{j-\nu_j}{L})}. \tag{163}$$

Due to this modification, it is enough to expand $\log \tilde{B}_{A,L}[\nu(q)]$ up to the linear terms in $\nu_j$ since higher orders will be of order $O(1/L)$, namely

$$\log \tilde{B}_{A,L}[\nu(q)] = \sum_{j=1}^{A} \nu_j \left( \frac{1}{j} - \frac{1}{L-j} - \frac{1}{L} \cot \frac{\pi j}{L} \right) + O(1/L). \tag{164}$$

Taking into account (153) we transform the sum into an integral

$$\log \tilde{B}_{A,L}[\nu(q)] = \int_{-\pi}^{q_A} \nu(q)\left(\frac{1}{q+\pi} - \frac{1}{\pi-q} + \frac{1}{2\pi}\tan\frac{q}{2}\right)dq. \tag{165}$$

The rest of the product can be evaluated in a similar manner. First, we identically transform

$$\prod_{j=1}^{A}\left(1-\frac{\nu_j}{j}\right)\left(1+\frac{\nu_j}{L-j}\right) = \Gamma\left[\begin{array}{c} A+1-\nu_1, L+\nu_A, L-A \\ A+1, 1-\nu_1, L, L-A+\nu_A \end{array}\right]\prod_{j=1}^{A}\frac{\left(1-\frac{\nu_j}{j}\right)\left(1+\frac{\nu_j}{L-j}\right)}{\left(1-\frac{\nu_1}{j}\right)\left(1+\frac{\nu_A}{L-j}\right)}. \tag{166}$$

The logarithm of the remaining product can be expanded only up to linear in $\nu$ terms to capture finite terms in $L \to \infty$ limit, namely

$$\log\prod_{j=1}^{A}\frac{\left(1-\frac{\nu_j}{j}\right)\left(1+\frac{\nu_j}{L-j}\right)}{\left(1-\frac{\nu_1}{j}\right)\left(1+\frac{\nu_A}{L-j}\right)} = -\int_{-\pi}^{q_A}dq\frac{\nu(q)-\nu(-\pi)}{q+\pi} + \int_{-\pi}^{q_A}dq\frac{\nu(q)-\nu_A}{\pi-q}. \tag{167}$$

Combining this result with (165) and using Stirling's formula we obtain the desired result (160). $\qquad\square$

**Remark 1.** For $A = L-1$, using Stirling's approximation for Gamma functions we obtain

$$\prod_{j=1}^{L-1}\frac{\sin\frac{\pi(j-\nu_j)}{L}}{\sin\frac{\pi j}{L}} \approx \frac{L^{\nu_+-\nu_-}}{\Gamma(1+\nu_+)\Gamma(1-\nu_-)}\exp\int_{-\pi}^{\pi}dq\left(\frac{\pi(\nu_+-\nu_-)+q(\nu_++\nu_-)}{q^2-\pi^2} + \frac{\nu(q)}{2}\tan\frac{q}{2}\right). \tag{168}$$

**Remark 2.** For $A \sim L$ and $L-A \sim L$ the prefactor can be simplified as

$$\Gamma\left[\begin{array}{c} A+1-\nu_1, L+\nu_A, L-A \\ A+1, 1-\nu_1, L, L-A+\nu_A \end{array}\right] = \frac{1}{L^{\nu_1}}\frac{1}{\Gamma(1-\nu_1)}\frac{1}{(A/L)^{\nu_1}(1-A/L)^{\nu_A}}. \tag{169}$$

The next lemma is a simple corollary of the previous one.

**Lemma B.3.** *The following asymptotic expression is valid as $L \to \infty$*

$$\mathcal{Z}_a \equiv \sin^2\frac{\pi\nu_a}{L}\prod_{j\neq a}^{L}\frac{\sin^2\frac{\pi(j-a-\nu_j)}{L}}{\sin^2\frac{\pi(j-a)}{L}} \approx L^{2\delta-2}\sin^2(\pi\nu_a)\Gamma\left[\begin{array}{c} a+\nu_a, L-a+1-\nu_a \\ a+\nu_+, L-a+1-\nu_- \end{array}\right]^2 e^{2F(q_a)}, \tag{170}$$

*where*

$$F(q_a) = \int_{q_a}^{\pi}\left(-\frac{\nu_+}{2\pi+q_a-q} + \frac{\nu_a}{q-q_a} - \frac{\nu(q)}{2}\cot\frac{q-q_a}{2}\right)dq$$

$$+ \int_{-\pi}^{q_a}\left(\frac{\nu_-}{2\pi+q-q_a} + \frac{\nu_a}{q-q_a} - \frac{\nu(q)}{2}\cot\frac{q-q_a}{2}\right)dq. \tag{171}$$

*Proof.* First, we identically present this product as

$$\mathcal{Z}_a = \sin^2\frac{\pi\nu_a}{L}\prod_{j=1}^{a-1}\frac{\sin^2\frac{\pi(j+\nu_{a-j})}{L}}{\sin^2\frac{\pi j}{L}}\prod_{j=1}^{L-a}\frac{\sin^2\frac{\pi(j-\nu_{j+a})}{L}}{\sin^2\frac{\pi j}{L}}. \tag{172}$$

Then using Lemma (B.2) and Stirling's formula we obtain

$$\mathcal{Z}_a \approx L^{2\delta-2} \sin^2(\pi \nu_a)\Gamma\left[\begin{array}{c} a + \nu_a, L - a + 1 - \nu_a \\ a + \nu_+, L - a + 1 - \nu_- \end{array}\right]^2 e^{2F(q_a)}, \tag{173}$$

with

$$F(q_a) = \int_{-\pi}^{-q_a} dq \left(-\frac{\nu_+}{\pi - q} + \frac{\nu(q_a)}{\pi + q} + \frac{\nu(q_a + q + \pi)}{2} \tan\frac{q}{2}\right) -$$
$$- \int_{-\pi}^{q_a} dq \left(-\frac{\nu_-}{\pi - q} + \frac{\nu(q_a)}{\pi + q} + \frac{\nu(q_a - q - \pi)}{2} \tan\frac{q}{2}\right). \tag{174}$$

Changing variables we obtain the desired statement. $\square$

Further, we proceed with double products.

**Lemma B.4.** *For $\delta \geq 0$ the following asymptotic expansion is valid in the limit $L \to \infty$*

$$Z \equiv \prod_{i=1}^{L}\prod_{j=1}^{i-1} \frac{\sin\frac{\pi}{L}(i - j - \nu_i + \nu_j)}{\sin\frac{\pi(i-j)}{L}} \approx \frac{\mathcal{A}}{L^{\delta^2/2}}, \tag{175}$$

*where the $L$ independent prefactor $\mathcal{A}$ reads*

$$\mathcal{A} = G(1 + \delta)(2\pi)^{-\frac{\delta(\delta+1)}{2}} \exp\left(\frac{\delta}{2} - \delta F(\pi) - \int_{-\pi}^{\pi} dq \int_{-\pi}^{\pi} dk \left[\frac{\nu(q) - \nu(k) - \delta(q - k)/(2\pi)}{4\sin\frac{q-k}{2}}\right]^2\right), \tag{176}$$

*with $F(\pi)$ is defined in Eq. (171) and $G(x)$ stands for Barnes G-function defined by the functional relation $G(x + 1) = \Gamma(x)G(x)$. Notice that function $\nu(q) - \delta q/2\pi$ has zero winding number so the integrals in the exponential are well defined.*

*Proof.* To find the thermodynamic limit of $Z$ we rewrite it as $Z = Y_1 Y_2 e^{R_\delta}$ with

$$Y_1 = \prod_{i=1}^{L}\prod_{j=1}^{i-1} \frac{\sin\frac{\pi}{L}(i - j - \nu_i + \nu_j)}{\sin\frac{\pi(i-j)}{L}} \frac{1 - \frac{i-j}{L}}{1 - \frac{i-j-(\nu_i-\nu_j)}{L}} e^{\frac{\nu_i-\nu_j}{L-i+j}} =$$
$$= \prod_{i=1}^{L}\prod_{j=1}^{i-1} \frac{\cos\frac{\pi(\nu_i-\nu_j)}{L}}{1 + \frac{\nu_i-\nu_j}{L(1-\frac{i-j}{L})}}\left(1 - \frac{\tan\frac{\pi(\nu_i-\nu_j)}{L}}{\tan\frac{\pi(i-j)}{L}}\right) e^{\frac{\nu_i-\nu_j}{L-i+j}}, \tag{177}$$

$$Y_2 = \prod_{i=1}^{L}\prod_{j=1}^{i-1}\left(1 + \frac{\delta}{L - i + j}\right) e^{-\frac{\delta}{L-i+j}} \prod_{i=1}^{L}\prod_{j=1}^{i-1}\left(1 + \frac{\nu_i - \nu_j - \delta}{L - i + j + \delta}\right) e^{-\frac{\nu_i-\nu_j-\delta}{L-i+j+\delta}}, \tag{178}$$

$$R_\delta = \sum_{i=1}^{L}\sum_{j=1}^{i-1}\left(\frac{\nu_i - \nu_j - \delta}{L - i + j + \delta} + \frac{\delta}{L - i + j} - \frac{\nu_i - \nu_j}{L - i + j}\right). \tag{179}$$

The factors are designed in such a way that terms $O(\nu^n)$ for $n > 2$ do not contribute in $L \to \infty$ case. In particular, we used that

$$\sum_{j=1}^{L-1} \cot\frac{\pi j}{L} = 0. \tag{180}$$

So keeping only quadratic terms we obtain

$$\log Y_1 = \sum_{i=1}^{L} \sum_{j=1}^{i-1} \frac{\pi^2}{2L^2} (v_i - v_j)^2 \left( \frac{1}{\pi^2} \left( 1 - \frac{i-j}{L} \right)^{-2} - \frac{1}{\sin^2 \frac{\pi(i-j)}{L}} \right) \tag{181}$$

and taking $L \to \infty$

$$\log Y_1 = \frac{1}{8} \int_{-\pi}^{\pi} dq \int_{-\pi}^{q} dk (v(q) - v(k))^2 \left( \frac{4}{(2\pi - q + k)^2} - \frac{1}{\sin^2 \frac{q-k}{2}} \right). \tag{182}$$

Similarly

$$\log \prod_{i=1}^{L} \prod_{j=1}^{i-1} \left( 1 + \frac{v_i - v_j - \delta}{L - i + j + \delta} \right) e^{-\frac{v_i - v_j - \delta}{L - i + j + \delta}} \approx -\frac{1}{2} \int_{-\pi}^{\pi} dq \int_{-\pi}^{q} dk \left( \frac{v(q) - v(k) - \delta}{2\pi - q + k} \right)^2. \tag{183}$$

The first part of the product in $Y_2$ (by grouping terms with the same $i - j$) can be presented as

$$W(\delta) \equiv \prod_{i=1}^{L} \prod_{j=1}^{i-1} \left( 1 + \frac{\delta}{L - i + j} \right) e^{-\frac{\delta}{L-i+j}} = \prod_{j=1}^{L-1} \left[ \left( 1 + \frac{\delta}{j} \right)^j e^{-\delta} \right]. \tag{184}$$

We consider an additional expression

$$W_0(\delta) \equiv \prod_{j=1}^{L-1} \left( 1 + \frac{\delta}{j} \right) = \frac{\Gamma(L + \delta)}{\Gamma(1 + \delta)\Gamma(L)}. \tag{185}$$

Differentiating it by $\delta$ we obtain

$$\frac{d \log W(\delta)}{d\delta} = -\delta \frac{d \log W_0(\delta)}{d\delta}. \tag{186}$$

For large $L$ we can approximate

$$W_0(\delta) \approx \frac{L^\delta}{\Gamma(1 + \delta)}. \tag{187}$$

Solving Eq. (186) with initial condition $\log W(\delta = 0) = 0$ we obtain

$$\log W(\delta) \approx -\frac{\delta^2}{2} \log L + \int_0^{\delta} z \frac{d \log \Gamma(1 + z)}{dz} dz. \tag{188}$$

Finally, let us find $L \to \infty$ expression for $R_\delta$ defined in Eq. (179). First we identically transform it into

$$R_\delta = \sum_{i=1}^{L} (v_i - v_L) S_{L-i+1} - \sum_{i=1}^{L} (v_i - v_1) S_i. \tag{189}$$

$$S_i = \sum_{j=i}^{L-1} \left( \frac{1}{j + \delta} - \frac{1}{j} \right) = \frac{d}{d\epsilon} \log \frac{\Gamma(L + \epsilon + \delta)\Gamma(i + \epsilon)}{\Gamma(L + \epsilon)\Gamma(i + \epsilon + \delta)} \Big|_{\epsilon=0}. \tag{190}$$

From the form of Eq. (189) one can conclude that as $L \to \infty$ the non-vanishing contributions to the sum will come from indices $i = O(L)$. Therefore, using Stirling's formula we can present $S_i$ as

$$S_i \approx \frac{d}{d\epsilon} \log \left( \frac{L + \epsilon}{i + \epsilon} \right)^\delta \Big|_{\epsilon=0} = \delta \left( \frac{1}{L} - \frac{1}{i} \right). \tag{191}$$

Therefore $R_\delta \approx \delta R$ with

$$
R = \lim_{L\to\infty} \left( \sum_{i=1}^{L} (\nu_i - \nu_L) \left( \frac{1}{L} - \frac{1}{L-i+1} \right) - \sum_{i=1}^{L-1} (\nu_i - \nu_1) \left( \frac{1}{L} - \frac{1}{i} \right) \right) =
$$
$$
= \int_{-\pi}^{\pi} dq (\nu(q) - \nu(\pi)) \left( \frac{1}{2\pi} - \frac{1}{\pi - q} \right) - \int_{-\pi}^{\pi} dq (\nu(q) - \nu(-\pi)) \left( \frac{1}{2\pi} - \frac{1}{\pi + q} \right). \quad (192)
$$

So far we have proved that

$$
Z \approx L^{-\delta^2/2} e^{C_\delta}, \quad (193)
$$

with

$$
C_\delta = \delta R + \int_0^\delta z \frac{d \log \Gamma(1+z)}{dz} dz + \frac{1}{2} \int_{-\pi}^{\pi} dq \int_{-\pi}^{q} dk \left( \frac{2\delta(\nu(q) - \nu(k)) - \delta^2}{(2\pi - q + k)^2} - \frac{(\nu(q) - \nu(k))^2}{4 \sin^2 \frac{q-k}{2}} \right). \quad (194)
$$

Further, we can use

$$
\int_0^\delta z \frac{d \log \Gamma(1+z)}{dz} dz = \frac{\delta(\delta+1)}{2} - \frac{\delta}{2} \log(2\pi) + \log G(1+\delta), \quad (195)
$$

where $G(x)$ is Barnes G-function. The final answer is obtained by tedious but straightforward manipulations with integrals. □

In the next lemma we address a similar double product for negative winding numbers $\delta < 0$.

**Lemma B.5.** *Let us define $\ell = L + \delta$, with $\delta < 0$, then the following asymptotic behavior is valid as $L \to \infty$ (here we still assume that $|\delta| \ll L$)*

$$
Z \equiv \prod_{i=1}^{\ell} \prod_{j=1}^{i-1} \frac{\sin \frac{\pi}{L}(i - j - \nu_i + \nu_j)}{\sin \frac{\pi(i-j)}{L}}
$$
$$
\approx \frac{L^{\delta^2/2} (2\pi)^{-(\delta^2+\delta)/2} e^{\delta/2}}{G(1-\delta)} \exp \left( -\int_{-\pi}^{\pi} dq \int_{-\pi}^{\pi} dk \left[ \frac{\nu(q) - \nu(k) - \delta(q-k)/(2\pi)}{4 \sin \frac{q-k}{2}} \right]^2 \right). \quad (196)
$$

*Proof.* We present this product as a ratio $Z = Z_1/Z_2$ with

$$
Z_1 = \prod_{i=1}^{\ell} \prod_{j=1}^{i-1} \frac{\sin \frac{\pi}{L}(i - j - \nu_i + \nu_j)}{\sin \frac{\pi(i-j)}{\ell}}, \qquad Z_2 = \prod_{i=1}^{\ell} \prod_{j=1}^{i-1} \frac{\sin \frac{\pi(i-j)}{L}}{\sin \frac{\pi(i-j)}{\ell}}. \quad (197)
$$

We can identically transform $Z_1$ as

$$
Z_1 = \prod_{i=1}^{\ell} \prod_{j=1}^{i-1} \frac{\sin \frac{\pi}{\ell}(i - j - [\nu_\delta]_i + [\nu_\delta]_j)}{\sin \frac{\pi(i-j)}{\ell}}, \quad (198)
$$

where $[\nu_\delta]_i = \nu_i(1 + \delta/L) - \delta i/L$. In thermodynamic limit this expression correspond to the following function

$$
\nu_\delta(q) = \nu(q) - \delta \frac{\pi + q}{2\pi}. \quad (199)
$$

This function has zero winding number, so applying the previous lemma, we obtain

$$Z_1 = \exp\left(-\int\limits_{-\pi}^{\pi} dq \int\limits_{-\pi}^{\pi} dk \left[\frac{v(q) - v(k) - \delta(q-k)/(2\pi)}{4\sin\frac{q-k}{2}}\right]^2\right). \tag{200}$$

Similarly, we can evaluate $Z_2$. We present it as

$$Z_2 = \prod_{i=1}^{\ell}\prod_{j=1}^{i-1} \frac{\sin\frac{\pi}{\ell}\left(i-j+\frac{\delta(i-j)}{L}\right)}{\sin\frac{\pi(i-j)}{\ell}}. \tag{201}$$

This corresponds to the positive phase shift $v(q) = -\delta q/(2\pi) = |\delta|q/(2\pi)$, and allows us to use previous lemma once again and obtain

$$Z_2 = \frac{G(1-\delta)(2\pi)^{(\delta^2+\delta)/2}e^{-\delta/2}}{\ell^{\delta^2/2}}. \tag{202}$$

$\square$

Here we used that $F(\pi) = -|\delta|\log(2\pi)$ for $v(q) = |\delta|q/(2\pi)$ (see Eq. (171)). Finally, Eqs. (200) and (202) immediately lead to the statement of the lemma.

## C Orthogonality catastrophe on the lattice

Here using results from Appendix B we evaluate the overlaps in Eq. (9).

### C.1 Winding number $\delta = 1$

For $\delta = 1$ there exist $L+1$ solutions of Eq. (6)

$$k_j = \frac{2\pi}{L}\left(-\frac{L+1}{2} + j - v_j\right), \qquad v_j = v(k_j), \qquad j = 1, 2, \ldots, L+1. \tag{203}$$

We use all of them in Eq. (9) and set $\mathbf{q} = \{q_1, \ldots q_L\}$ with

$$q_j = \frac{2\pi}{L}\left(-\frac{L+1}{2} + j\right), \qquad j = 1, 2, \ldots, L. \tag{204}$$

To evaluate Eq. (9) in thermodynamic limit $L \to \infty$, we first we transform identically the determinant (10) as

$$(\det D)^2 = \prod_{i>j}^{L} \frac{\sin^2\frac{k_i-k_j}{2}}{\sin^2\frac{q_i-q_j}{2}} \times \prod_{j=1}^{L} \frac{\sin^2\frac{k_{L+1}-k_j}{2}}{\sin^2\frac{k_{L+1}-q_j}{2}} \times \prod_{i=1}^{L} \frac{\prod_{j\neq i}^{L}\sin^2\frac{q_i-q_j}{2}}{\prod_{j=1}^{L}\sin^2\frac{k_i-q_j}{2}}. \tag{205}$$

We analyze this expression term by term. The last product can be written down using Eqs. (19) and (20) as

$$\frac{\prod_{j\neq i}^{L}\sin^2\frac{q_i-q_j}{2}}{\prod_{j=1}^{L}\sin^2\frac{k_i-q_j}{2}} = \frac{1}{\sin^2\frac{\pi v_i}{L}}\prod_{j=1}^{L-1}\frac{\sin^2\frac{\pi j}{L}}{\sin^2\frac{\pi(j-v_i)}{L}} = \frac{L^2}{\sin^2\pi v_i}. \tag{206}$$

In the last step, we used Lemma (B.1). The next product can be evaluated employing similar transformations and using Lemma (B.2), namely

$$\prod_{j=1}^{L} \frac{\sin^2 \frac{k_{L+1}-k_j}{2}}{\sin^2 \frac{k_{L+1}-q_j}{2}} = \frac{\sin^2 \frac{\pi\delta}{L}}{\sin^2 \frac{\pi\nu_+}{L}} \prod_{j=1}^{L-1} \frac{\sin^2 \frac{\pi}{L}\left(j - \nu_+ + \nu_{L+1-j}\right)}{\sin^2 \frac{\pi j}{L}} \prod_{j=1}^{L-1} \frac{\sin^2 \frac{\pi j}{L}}{\sin^2 \frac{\pi(j-\nu_+)}{L}}$$

$$\approx \frac{\pi^2 L^2}{\sin^2 \pi\nu_+} \exp\left(\int_{-\pi}^{\pi} dq f_1(q)\right), \tag{207}$$

where $\nu_+ = \nu_{L+1}$ and $\delta = \nu_+ - \nu_1 = \nu(\pi) - \nu(-\pi) = 1$ and

$$f_1(q) = \frac{2}{q-\pi} + (\nu(\pi) - \nu(-q)) \tan \frac{q}{2}. \tag{208}$$

Notice that

$$\int_{-\pi}^{\pi} dq f_1(q) = 2F(\pi) = 2F(-\pi), \tag{209}$$

with $F(q)$ defined in Eq. (171). Contrary to the expression (206), Eq. (207) is asymptotic as $L \to \infty$. Finally, the first double product in Eq. (205) can be evaluated using Lemma (B.4).

$$\prod_{i>j}^{L} \frac{\sin^2 \frac{k_i-k_j}{2}}{\sin^2 \frac{q_i-q_j}{2}} = \prod_{i=1}^{L} \prod_{j=1}^{i-1} \frac{\sin^2 \frac{\pi}{L}(i-j-\nu_i+\nu_j)}{\sin^2 \frac{\pi(i-j)}{L}} \approx \frac{\mathcal{A}^2}{L}. \tag{210}$$

Where $A$ is defined in Eq. (176). The rest of the product in Eq. (9) can be evaluated for generic $\delta$

$$\prod_{i=1}^{L+1}\left(1 + \frac{2\pi}{L}\nu'(k_i)\right) \approx \exp\left(\int_{-\pi}^{\pi} \nu'(q)dq\right) = e^{\delta}, \tag{211}$$

$$\prod_{i=1}^{L} e^{g(k_i)-g(q_i)} \approx \exp\left(-\int_{-\pi}^{\pi} g'(q)\nu(q)dq\right) = \exp\left(2\pi i \int_{-\pi}^{\pi} \frac{\nu(q)\nu'(q)}{e^{2\pi i \nu(q)}-1}dq\right) = \left(1 - e^{-2\pi i \nu_+}\right)^{\delta}, \tag{212}$$

where in the last part we have used the relation between $g(q)$ and $\nu(q)$ Eq. (15) and assumed that $\nu(k)$ has a non-vanishing imaginary part. Combining all factors together in Eq. (9) we obtain

$$|\langle \mathbf{k}|\mathbf{q}\rangle|^2 = 4\pi^2 \mathcal{A}^2 e^{2F(\pi)-1} = \exp\left(-\frac{1}{2}\int_{-\pi}^{\pi} dq \int_{-\pi}^{\pi} dk \left[\frac{\nu(q)-\nu(k)-(q-k)/2\pi}{2\sin\frac{q-k}{2}}\right]^2\right). \tag{213}$$

## C.2   Winding number $\delta = 2$

For $\delta > 1$ computation of the overlaps goes in the similar manner as in the previous section. Namely, first we consider overlap with the set $\tilde{\mathbf{k}} = k_1, \dots k_{L+1}$ with $k_j$ defined in Eq. (203). There instead of Eq. (207) we will have

$$\prod_{j=1}^{L} \frac{\sin^2 \frac{k_{L+1}-k_j}{2}}{\sin^2 \frac{k_{L+1}-q_j}{2}} = \frac{L^2}{\sin^2(\pi\nu_+)} \frac{\pi^2 L^{2\delta-2}}{\Gamma(\delta)^2} e^{2F(\pi)}, \tag{214}$$

with $F(\pi)$ defined in Eq. (171). Further, Eq. (210) we will replaced accordingly to Lemma (B.4)

$$\prod_{i>j}^{L} \frac{\sin^2 \frac{k_i - k_j}{2}}{\sin^2 \frac{q_i - q_j}{2}} = \prod_{i=1}^{L} \prod_{j=1}^{i-1} \frac{\sin^2 \frac{\pi}{L}(i - j - \nu_i + \nu_j)}{\sin^2 \frac{\pi(i-j)}{L}} \approx \frac{\mathcal{A}^2}{L^{\delta^2}}. \tag{215}$$

Taking into account Eqs. (211) and (212), for the corresponding function $g(k)$ (see Eq. (55)) we find the thermodynamic form for the overlap

$$|\langle \tilde{\mathbf{k}} | \mathbf{q} \rangle|^2 = \frac{G(\delta)^2}{L^{(\delta-1)^2}} \frac{(1 - e^{2\pi i \nu_+})^{\delta-1}}{(2\pi)^{(\delta-1)(\delta+2)}} e^{-2F(\pi)(\delta-1)} \exp\left( -\frac{1}{2} \int_{-\pi}^{\pi} dq \int_{-\pi}^{\pi} dk \left[ \frac{\nu(q) - \nu(k) - \delta(q-k)/2\pi}{2 \sin \frac{q-k}{2}} \right]^2 \right). \tag{216}$$

The overlaps for other sets $\mathbf{k}$ can be obtained from this one. We further focus on $\delta = 2$, in this case there are exacly $L + 2$ sets $\mathbf{k}$ parametrized by the omission of one of the solutions of Eq. (6), namely

$$\mathbf{k}^{(a)} = \{k_1, \ldots, k_{a-1}, k_{a+1}, \ldots, k_{L+2}\}, \qquad a = 1, 2, \ldots, L + 2. \tag{217}$$

With this notations $\mathbf{k}^{(L+2)} = \tilde{\mathbf{k}}$. Now let us consider ratio of the excited overlap

$$\frac{|\langle \mathbf{k}^{(a)} | \mathbf{q} \rangle|^2}{|\langle \tilde{\mathbf{k}} | \mathbf{q} \rangle|^2} = e^{g(\pi) - g(k_a)} \frac{\prod_{j=1}^{L+1} \sin^2 \frac{k_{L+2} - k_j}{2}}{\prod_{j \neq a}^{L+2} \sin^2 \frac{k_a - k_j}{2}} = e^{g(\pi) - g(k_a)} \frac{\sin^2 \frac{\pi}{L} \sin^2 \frac{2\pi}{L} \prod_{j=1}^{L-1} \sin^2 \frac{\pi}{L}(j - \nu_{j+2} + \nu_+)}{\prod_{j=1}^{a-1} \sin^2 \frac{\pi(j - \nu_a + \nu_{a-j})}{L} \prod_{j=1}^{L+2-a} \sin^2 \frac{\pi(j - \nu_{j+a} + \nu_a)}{L}}. \tag{218}$$

Using Lemma (B.2) for $a \sim L$, $L - a \sim L$ we obtain

$$\frac{|\langle \mathbf{k}^{(a)} | \mathbf{q} \rangle|^2}{|\langle \tilde{\mathbf{k}} | \mathbf{q} \rangle|^2} = (2\pi)^4 \exp\left[ 2F(\pi) + \int_{-\pi}^{\pi} \left( \nu(q) - \frac{q}{\pi} \right) \cot \frac{q - k_a}{2} dq \right]. \tag{219}$$

Combining this result with Eq. (216) for $\delta = 2$, the overlap can be written as

$$|\langle \mathbf{k}^{(a)} | \mathbf{q} \rangle|^2 = -\frac{e^{2\pi i \nu(k)} - 1}{L} \exp\left[ \int_{-\pi}^{\pi} \left( \nu(q) - \frac{q}{\pi} \right) \cot \frac{q - k_a}{2} dq - \frac{1}{2} \int_{-\pi}^{\pi} dq \int_{-\pi}^{\pi} dk \left( \frac{\nu(q) - \nu(k) - (q-k)/\pi}{2 \sin \frac{q-k}{2}} \right)^2 \right]. \tag{220}$$

## C.3 Winding number $\delta = 0$

Let us study thermodynamic limit of the overlap (9) in the case $N = L - 1$, which is especially useful for $\delta = 0$. Below, however, for the sake of generality, we will keep $\delta \geq 0$. Our goal is to evaluate $Z_a$ defined via

$$|\langle \mathbf{k} | \mathbf{q}^{(a)} \rangle|^2 \equiv e^{g(q_a)} Z_a. \tag{221}$$

We use notations (20) and (19) to label the momenta and Eq. (23) for $\mathbf{q}^{(a)}$. Let $\det D^{(a)}$ be the determinant in Eq. (10) that corresponds to the set $\mathbf{q}^{(a)}$. It explicitly reads as

$$\det D^{(a)} = \frac{\prod_{i>j}^{L} \sin \frac{k_i - k_j}{2} \prod_{\substack{i>j \\ i,j \neq a}}^{L} \sin \frac{q_j - q_i}{2}}{\prod_{i=1}^{L} \prod_{\substack{j=1 \\ j \neq a}}^{L} \sin \frac{k_i - q_j}{2}}. \tag{222}$$

We can present it identically as

$$\prod_{i=1}^{L}\left(\frac{\sin \pi \nu_i}{L}\right)^2 (\det D^{(a)})^2 = \prod_{i=1}^{L}\prod_{j=1}^{i-1}\frac{\sin^2 \frac{k_i-k_j}{2}}{\sin^2 \frac{q_i-q_j}{2}} \times \prod_{i=1}^{L}\left(\frac{\sin \pi \nu_i}{L}\right)^2 \frac{\prod_{j\neq i}^{L}\sin^2 \frac{q_i-q_j}{2}}{\prod_{j=1}^{L}\sin^2 \frac{k_i-q_j}{2}}$$

$$\times \sin^2 \frac{\pi \nu_a}{L} \prod_{j\neq a}\frac{\sin^2 \frac{k_j-q_a}{2}}{\sin^2 \frac{q_j-q_a}{2}}. \tag{223}$$

The last part of this product is nothing but $\mathcal{Z}_a$ in Eq. (170), the middle part is equal to 1 due to to Lemma (B.1), while the first part can evaluated with Lemma (B.4) and gives $\mathcal{A}^2/L^{\delta^2}$. Overall we have[4]

$$\prod_{i=1}^{L}\left(\frac{\sin \pi \nu_i}{L}\right)^2 (\det D^{(a)})^2 \approx \frac{\mathcal{A}^2 e^{2F(q_a)}}{L^{\delta^2-2\delta+2}}\sin^2(\pi \nu_a)\left[\frac{\Gamma(L-a+1-\nu_a)\Gamma(a+\nu_a)}{\Gamma(L-a+1-\nu_-)\Gamma(a+\nu_+)}\right]^2, \tag{224}$$

where $F(q_a)$ is given by Eq. (171).

Taking into account Eqs. (211) and (212), we obtain

$$Z_a = -4(1-e^{-2\pi i \nu_+})^{\delta}\frac{\mathcal{A}^2 e^{2F(q_a)-\delta}}{L^{(\delta-1)^2}}\sin^2(\pi \nu_a)\left[\frac{\Gamma(L-a+1-\nu_a)\Gamma(a+\nu_a)}{\Gamma(L-a+1-\nu_-)\Gamma(a+\nu_+)}\right]^2. \tag{225}$$

For $\delta = 0$ we can rewrite this expression as

$$Z_a = \frac{A[q_a]}{L}\left[\frac{\Gamma(L-a+1-\nu_a)\Gamma(a+\nu_a)}{\Gamma(L-a+1-\nu_+)\Gamma(a+\nu_+)}\right]^2\left(\frac{\pi+q_a}{\pi-q_a}\right)^{2\nu_+-2\nu_a}, \tag{226}$$

$$A[q_a] = -4\sin^2(\pi \nu_a)\exp\left(-\frac{1}{2}\int_{-\pi}^{\pi}dq\int_{-\pi}^{\pi}dk\left[\frac{\nu(q)-\nu(k)}{2\sin \frac{q-k}{2}}\right]^2 - \int_{-\pi}^{\pi}\nu(q)\cot \frac{q-q_a}{2}dq\right), \tag{227}$$

where the integral is understood as the principal value.

For $\delta = 1$ we can rewrite this expression as

$$Z_a = 4\sin^2(\pi \nu_a)(e^{-2\pi i \nu_+}-1)\mathcal{A}^2 e^{2F(q_a)-1}\left[\frac{\Gamma(L-a+1-\nu_a)\Gamma(a+\nu_a)}{\Gamma(L-a+2-\nu_+)\Gamma(a+\nu_+)}\right]^2. \tag{228}$$

Using expression (209) and (176) we obtain

$$Z_a = \frac{\sin^2(\pi \nu_a)}{\pi^2}(e^{-2\pi i \nu_+}-1)|\langle \mathbf{k}|\mathbf{q}\rangle|^2 e^{2F(q_a)-2F(\pi)}\left[\frac{\Gamma(L-a+1-\nu_a)\Gamma(a+\nu_a)}{\Gamma(L-a+2-\nu_+)\Gamma(a+\nu_+)}\right]^2, \tag{229}$$

with $|\langle \mathbf{k}|\mathbf{q}\rangle|^2$ given by Eq. (213).

## C.4 Negative winding number $\delta < 0$

Following Sec. (3.3) we fix $\delta = 1-n$ with $n \in \mathbb{Z}_{\geq}$, $\ell = L+\delta$, the set $\mathbf{k} = \{k_1, \ldots k_\ell\}$ is given as

$$k_i = \frac{2\pi}{L}\left(-\frac{L+1}{2}+i-\nu_i\right), \qquad i = 1, 2, \ldots \ell, \tag{230}$$

---

[4]Recall that $\nu_a \equiv \nu(q_a)$.

the set $\mathbf{q}^{a_1,\ldots a_n}$ is obtained from the complete set $\mathbf{q}$ in Eq. (204) by the omission of the "particle" at position $q_{a_i}$

$$\mathbf{q}^{a_1,\ldots a_n} = \{q_1,\ldots \hat{q}_{a_1},\ldots \hat{q}_{a_n},\ldots q_L\}. \tag{231}$$

The determinant (10) in (9) after certain restructuring of the factors and employing Lemma (B.1) reads

$$\prod_{i=1}^{\ell}\left(\frac{\sin \pi \nu_i}{L}\right)^2 (\det D)^2 = \prod_{i=1}^{\ell} \frac{\prod_{j=1}^{L} \sin^2 \frac{k_i-q_j}{2}}{\prod_{j\neq i}^{L} \sin^2 \frac{q_i-q_j}{2}} \times \frac{\prod_{i>j}^{\ell} \sin^2 \frac{k_i-k_j}{2} \prod_{\substack{i>j \\ i,j\neq a_1,\ldots,a_n}}^{L} \sin^2 \frac{q_j-q_i}{2}}{\prod_{i=1}^{\ell} \prod_{\substack{j=1 \\ j\neq a_1,\ldots,a_n}}^{L} \sin^2 \frac{k_i-q_j}{2}}$$

$$= \frac{\prod_{i>j}^{\ell} \sin^2 \frac{k_i-k_j}{2}}{\prod_{i>j}^{\ell} \sin^2 \frac{q_i-q_j}{2}} \times \frac{\prod_{i>j}^{\ell} \sin^2 \frac{q_i-q_j}{2} \prod_{i>j}^{L} \sin^2 \frac{q_j-q_i}{2}}{\prod_{i=1}^{\ell} \prod_{\substack{j=1 \\ j\neq i}}^{L} \sin^2 \frac{q_i-q_j}{2}} \times \prod_{i>j}^{n} \sin^2 \frac{q_{a_i}-q_{a_j}}{2} \times \prod_{i=1}^{n} \tilde{\mathcal{Z}}_{a_i}, \tag{232}$$

with

$$\tilde{\mathcal{Z}}_a = \frac{\prod_{i=1}^{\ell} \sin^2 \frac{k_i-q_a}{2}}{\prod_{i\neq a}^{L} \sin^2 \frac{q_i-q_a}{2}}. \tag{233}$$

The first factor in this expression can be evaluated via Lemma (B.5)

$$\frac{\prod_{i>j}^{\ell} \sin^2 \frac{k_i-k_j}{2}}{\prod_{i>j}^{\ell} \sin^2 \frac{q_i-q_j}{2}} = \frac{L^{\delta^2}(2\pi)^{-(\delta^2+\delta)}e^{\delta}}{G(1-\delta)^2} \exp\left(-\frac{1}{2}\int_{-\pi}^{\pi} dq \int_{-\pi}^{\pi} dk \left[\frac{\nu(q)-\nu(k)-\delta(q-k)/(2\pi)}{2\sin \frac{q-k}{2}}\right]^2\right). \tag{234}$$

The second factor reads

$$\frac{\prod_{i>j}^{\ell} \sin^2 \frac{q_i-q_j}{2} \prod_{i>j}^{L} \sin^2 \frac{q_j-q_i}{2}}{\prod_{i=1}^{\ell} \prod_{\substack{j=1 \\ j\neq i}}^{L} \sin^2 \frac{q_i-q_j}{2}} = \prod_{i>j>\ell}^{L} \sin^2 \frac{q_i-q_j}{2} = \prod_{i=1}^{n-1} \prod_{j=1}^{i-1} \sin^2 \frac{\pi(i-j)}{L} \approx \left(\frac{\pi}{L}\right)^{(n-2)(n-1)} \prod_{i=1}^{n-1} \prod_{j=1}^{i-1} (i-j)^2$$

$$= \left(\frac{\pi}{L}\right)^{(n-2)(n-1)} \prod_{i=1}^{n-1} \prod_{j=1}^{i-1} j^2 = \left(\frac{\pi}{L}\right)^{(n-2)(n-1)} \prod_{i=1}^{n-1} \Gamma(i)^2 = \left(\frac{\pi}{L}\right)^{(n-2)(n-1)} G(n)^2 = \left(\frac{\pi}{L}\right)^{\delta(\delta+1)} G(1-\delta)^2. \tag{235}$$

We evaluate $\tilde{\mathcal{Z}}_a$ in Eq. (233) for $a \sim L$ and $L-a \sim L$. We complete $\tilde{\mathcal{Z}}_a$ to the full product $\mathcal{Z}_a$ in Eq. (170) and approximate it as

$$\tilde{\mathcal{Z}}_a = \frac{\mathcal{Z}_a}{\prod_{j=\ell+1}^{L} \sin^2 \frac{\pi(j-a-\nu_j)}{L}} \approx \frac{\mathcal{Z}_a}{\left(\cos \frac{q_a}{2}\right)^{2|\delta|}}. \tag{236}$$

To approximate further $\mathcal{Z}_a$ in Eq. (170) we notice that

$$
L^{\delta}\Gamma\left[\begin{array}{c} a + \nu_a, \, L-a+1-\nu_a \\ a + \nu_+, \, L-a+1-\nu_- \end{array}\right] \approx \left(\frac{a}{L}\right)^{\nu_a - \nu_+}\left(1-\frac{a}{L}\right)^{\nu_- - \nu_a} = \left(\frac{\pi+q_a}{2\pi}\right)^{\nu_a - \nu_+}\left(\frac{\pi-q_a}{2\pi}\right)^{\nu_- - \nu_a}.
$$
(237)

Further, we can simplify $F(q_a)$ using that in the principal value

$$
\fint_{-\pi}^{\pi} dq \, q \cot\frac{q-q_a}{2} = 4\pi \log\left|2\cos\frac{q_a}{2}\right|.
$$
(238)

So thermodynamic limit for $\tilde{\mathcal{Z}}_a$ reads

$$
\tilde{\mathcal{Z}}_a = 4^{|\delta|}\frac{\sin^2(\pi\nu_a)}{L^2}\exp\left[-\fint_{-\pi}^{\pi} dq\left(\nu(q)-\delta\frac{q}{2\pi}\right)\cot\frac{q-q_a}{2}\right].
$$
(239)

The remaining factors in Eq. (9) can be evaluated with the help of Eqs. (211) and (212)

$$
\frac{\prod_{i=1}^{\ell} e^{g(k_i)} \prod_{q_i \in \mathbf{q}^{a_1,\dots a_n}} e^{-g(q_i)}}{\prod_{i=1}^{\ell}\left(1+\frac{2\pi}{L}\nu'(k_i)\right)} = \frac{\prod_{i=1}^{\ell} e^{g(k_i)-g(q_i)} \prod_{i=\ell+1}^{L} e^{-g(q_i)} \prod_{i=1}^{n} e^{g(q_{a_i})}}{\prod_{i=1}^{\ell}\left(1+\frac{2\pi}{L}\nu'(k_i)\right)} = (-1)^{\delta} e^{-\delta} \prod_{i=1}^{n} e^{g(q_{a_i})}.
$$
(240)

Finally, the overlap (9) in the thermodynamic limit can be written as

$$
|\langle\mathbf{k}|\mathbf{q}^{a_1,\dots a_n}\rangle|^2 = \exp\left(-\frac{1}{2}\int_{-\pi}^{\pi} dq\int_{-\pi}^{\pi} dk\left[\frac{\nu(q)-\nu(k)-\delta(q-k)/(2\pi)}{2\sin\frac{q-k}{2}}\right]^2\right) \prod_{i>j}^{n}\left(2\sin\frac{q_{a_i}-q_{a_j}}{2}\right)^2 \prod_{i=1}^{n}\mathcal{Y}_{a_i}
$$
(241)

with

$$
\mathcal{Y}_a = -4\frac{\sin^2(\pi\nu_a)}{L}\exp\left[g(q_a)-\fint_{-\pi}^{\pi} dq\left(\nu(q)-\delta\frac{q}{2\pi}\right)\cot\frac{q-q_a}{2}\right].
$$
(242)

## C.5 Overlaps for $\tau_0$

Now let us consider how overlaps defined in Eq. (99) scale with the system size for $\delta \leq 0$. Similarly, to the previous sections we can present solutions of Eq. (98) as

$$
p_i = \frac{2\pi}{L}\left(-\frac{L+1}{2}+j-\omega_j\right), \qquad j = 1,\dots,\ell = L+\delta.
$$
(243)

We use maximally allows set for $\mathbf{p}$, namely

$$
\mathbf{p} = \{p_1,\dots p_\ell\}
$$
(244)

and states $\mathbf{q}$ are parametrized by the set of $n = |\delta|$ holes as previously

$$
\mathbf{q}^{a_1,\dots a_n} = \{q_1,\dots\hat{q}_{a_1},\dots\hat{q}_{a_n},\dots q_L\}.
$$
(245)

Similar to Eq. (232) using Lemma (B.1) the overlap (99) can be presented as

$$
|\langle \mathbf{p}|\mathbf{q}^{a_1,\dots a_n}\rangle|^2 = \frac{\prod\limits_{i=1}^{\ell} e^{g(p_i)-g(q_i)} \prod\limits_{i=1}^{n} e^{g(q_{a_i})-g(\pi)} \prod\limits_{i>j}^{\ell} \sin^2 \frac{p_i-p_j}{2}}{\prod\limits_{i=1}^{\ell}\left(1+\frac{2\pi}{L}\omega'(p_i)\right) \quad \prod\limits_{i>j}^{\ell} \sin^2 \frac{q_i-q_j}{2}}
$$

$$
\times \frac{\prod\limits_{i>j}^{\ell} \sin^2 \frac{q_i-q_j}{2} \prod\limits_{i>j}^{L} \sin^2 \frac{q_j-q_i}{2}}{\prod\limits_{i=1}^{\ell}\prod\limits_{\substack{j=1 \\ j\neq i}}^{L} \sin^2 \frac{q_i-q_j}{2}} \times \prod\limits_{i>j}^{n} \sin^2 \frac{q_{a_i}-q_{a_j}}{2} \times \prod\limits_{k=1}^{n} \frac{\prod\limits_{i=1}^{\ell} \sin^2 \frac{p_i-q_{a_k}}{2}}{\prod\limits_{i\neq a_k}^{L} \sin^2 \frac{q_i-q_{a_k}}{2}}. \quad (246)
$$

This way, using formulas from the previous subsection (C.4), we see that overlap $|\langle \mathbf{p}|\mathbf{q}^{a_1,\dots a_n}\rangle|^2$ is identical to Eqs. (241), (242), upon the identification $\nu \to \omega$ and $\delta$ to be changed from by $\nu(\pi)-\nu(-\pi) \to \omega(\pi)-\omega(-\pi)$.

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
