# Peer review of "Effective free-fermionic form factors and the XY spin chain"

_SciPost Physics, doi:SciPost Phys. 10, 070 (2021)_

## Round 3 · Referee Report · Anonymous · 2021-2-16

Strengths

see report

Weaknesses

see report

Report

The paper "Effective free-fermionic form factors and the XY spin chain`` by O. Gamayun, N. Iorgov, Yu. Zhuravlev applies techniques of form factor summations in generalised free fermionic models so as to reproduce the
large distance asymptotics of the thermal spin spin correlation functions in the XY spin chain.

In overall, the paper is well written and easy to read. The scope of techniques used in the paper is also up to date in respect to the ongoing research relative to correlation functions in interacting integrable models. It is a bit of a pity that the authors did not connect, even be it on phenomenological grounds, the specific form factor expansion they consider, with some setting appearing in the interacting case. Also, I am unsure how general is the scope of applicability of the "thermal" summations obtained by the authors in that the range of summations they carry out in the form factor expansions seems rather particular: a very limited amount of intermediate states is only considered. It would be interesting if they could develop on how they would expect to cope with a bona fide thermal expansion where one sums up over all states' form factors weighted by appropriate Boltzmann factors, e.g. as done in ref. 47-48. Still, that side remark does not prevent me from recommending the paper for publication. I would ask the authors to take into account the below remarks.

Requested changes

1) The authors should clarify what they mean by the sentence "as the form-factors (matrix elements of the physical operators) decay exponentially with the systems size contrary to the power-law decays at zero temperatures". Indeed, the form factors arising in the thermal expansion of a correlator do contain, as a subset, the form factors contributing to the zero temperature expansion, which, as they write, decay as a power-law of the system size.

2) It would be useful if the authors explain explicitly around (5) that the parameter N will be specialised to various values throughout the paper.

3) (17) should be $\delta V_{\nu}$

4) The authors should cite the work V.E.~Korepin and N.A.~Slavnov, The time dependent correlation function of an impenetrable Bose gas as a Fredholm minor I, Comm. Math.Phys. {\bf 129}, (1990), 103-113. They should make clear in the paper that the form factor summation method they use was pioneered there.

5) The authors write "In this case we see that even though the difference between V and S is exponentially small, it cannot be neglected, contrary". It is not enough that the kernels are pointwise exponentially small. There must be some uniformness on top of that. For instance an exponentially small trace class norm estimate for the perturbation along with trace class estimates for the unperturbed operator. Thus the property is not so astonishing. The authors should thus find some explanation of the sort.

6) It would be interesting if the authors developed a bit more on the time dependent case.

7) The authors should give more hints on why they think that they can get through their method "the full asymptotic expansion at large distances.".

  • validity: good
  • significance: good
  • originality: good
  • clarity: high
  • formatting: excellent
  • grammar: good

Author:  Oleksandr Gamayun  on 2021-03-11  [id 1299]

(in reply to Report 1 on 2021-02-16)

We are grateful to the Referee for their assessment of our work and for their comments. Let us address each point in turn.

1) The authors should clarify what they mean by the sentence "as the form-factors (matrix elements of the physical operators) decay exponentially with the systems size contrary to the power-law decays at zero temperatures". Indeed, the form factors arising in the thermal expansion of a correlator do contain, as a subset, the form factors contributing to the zero temperature expansion, which, as they write, decay as a power-law of the system size.

The form factor depends on two sets of vectors (bra and ket, k and q). In the summation in Gibbs ensemble the measure of the bra-states (k) that corresponds to zero temperature is zero. The most typical (representative) is the state with the quantum numbers spread accordingly to the equilibrium density. We assume that we can fix the set k to be this representative state (see section I.8 in [57]), then the other set runs over the whole Hilbert space. The power-law decay is possible only when the representative state corresponds to the zero temperature. For the finite temperature, all form factors decay exponentially with the system size even if the quantum numbers (integers n_i) for k and q coincide. That is why the straightforward bona fide thermal summation is complicated (even numerically) due to the huge number of form factors to be taken into account. To deal with the problem we introduce the effective form factors for which the number of different states in the series is significantly reduced.

2) It would be useful if the authors explain explicitly around (5) that the parameter N will be specialised to various values throughout the paper.

We have added the corresponding discussion.

3) (17) should be δVν

Fixed.

4) The authors should cite the work V.E.~Korepin and N.A.~Slavnov, The time dependent correlation function of an impenetrable Bose gas as a Fredholm minor I, Comm. Math.Phys. {\bf 129}, (1990), 103-113. They should make clear in the paper that the form factor summation method they use was pioneered there.

We have added the citation and the appropriate remarks.

5) The authors write "In this case we see that even though the difference between V and S is exponentially small, it cannot be neglected, contrary". It is not enough that the kernels are pointwise exponentially small. There must be some uniformness on top of that. For instance an exponentially small trace class norm estimate for the perturbation along with trace class estimates for the unperturbed operator. Thus the property is not so astonishing. The authors should thus find some explanation of the sort.

The discarding exponentially small terms under the Fredholm determinant is a delicate thing as we discuss in Sec. 3.4. The available estimation Eq. 4.2 in [68] is too rough to be applied for our case. It actually states that this correction cannot be discarded even in the case when it works. As a rule of thumb, we found out that the exponentially small terms can be safely neglected if they vanish faster than the total correlator. However, the full mathematical analysis is not at hand at the moment. Luckily we could check our conjectures with numerics.

6) It would be interesting if the authors developed a bit more on the time dependent case.

We expect that on the level of the form factor summation the procedure stays the same. The crucial difference will be to find the corresponding effective \nu, due to the possible saddle point in E(q). The meaningful discussion, however, has to be postponed to a separate publication.

7) The authors should give more hints on why they think that they can get through their method "the full asymptotic expansion at large distances.".

Since the Fredholm determinant with the modified kernel equals the elementary functions explicitly (without any corrections). We expect that we can obtain estimates not only for the determinant but also for the resolvent and thus extract the full asymptotic series. The details of this program are still under development.

---

## Round 3 · Referee Report · Anonymous · 2021-2-17

Strengths

1- the idea of replacing/approximating the overlaps by free overlaps that make the calculations tractable is interesting

2-the idea that for an appropriate number of particles one can express a Fredholm determinant as a sum of only one or a few overlaps is interesting

Weaknesses

1-The method takes as an input a function $\nu$. But this function itself is determined from the known results on XY correlations.

Report

The authors study the problem of computing correlation functions in integrable models within finite entropy states, which is relevant in a lot of situations such as finite temperature equilibrium. It is a timely and difficult problem, since these correlations involve a very large number of states.

They define a function $\tau(x)$ as sum of "abstract" overlaps, that they choose to be of Cauchy form and that depend on a function $\nu$ that is to be chosen. As the overlaps are Cauchy, the sum can be computed and expressed in terms of Fredholm determinants. Then they observe that for a particular number of particles the sum reduces to only one or a few states, which makes it directly computable (or its asymptotics at large $x$). They now notice that if they neglect a term in this Fredholm det and if they choose appropriately the function $\nu$, their function $\tau(x)$ matches the spin correlations in XY that has been calculated previously differently. They conclude that they can compute the large $x$ behaviour of the spin correlation in the XY model.

I find the paper interesting but also a bit misleading. An essential point of their reasoning is to choose the function $\nu$ so that $\tau$ matches the known result for the spin correlation in XY. But if we didn't know this result, we could not relate $\tau$ to XY, and we would not know how to choose $\nu$. We would not even know the answer can be expressed as a Fredholm. The correlation length itself is in fact essentially fixed by this choice of $\nu$ deduced from known result. For this reason the sentence in introduction "it allows to get exact answers for spin-spin correlation functions in terms of Toeplitz or Fredholm determinants" is not quite true in my opinion: first this representation is not exact (there is a piece neglected in the determinant), and second it requires to know the result to choose the appropriate $\nu$, so it cannot be said to be a derivation. All in all, the paper provides a method to compute the asymptotics of the Fredholm determinant. This method is interesting and original, but it does not derive this Fredholm determinant. This should be mentioned clearly at the places where the results are presented/summarized, otherwise it is misleading.

Some other remarks:
- The neglected piece in the Fredholm det is said to vanish exponentially with x. But at finite temperature, so do the correlations. It should complicate computation of the next orders.
- The calculation of the $\tau$ function was done by Korepin and Slavnov (Commun Math Phys 129, 103–113(1990)) and should be cited.

The paper is interesting for several points, so I recommend publication. But the necessity for choosing $\nu$ should be discussed/mentioned, and for this reason it should be made clear that it is not an alternative derivation of XY correlations.

Requested changes

1- discuss the necessity of choosing $\nu$ from the known result on XY correlations, and correct corresponding affirmations in the introduction (5th paragraph) conclusion (1st paragraph) and abstract
2- cite the reference mentioned in the report

  • validity: good
  • significance: good
  • originality: high
  • clarity: good
  • formatting: good
  • grammar: good

Author:  Oleksandr Gamayun  on 2021-03-11  [id 1298]

(in reply to Report 2 on 2021-02-17)

We thank the Referee for the evaluation of our work and their comments. Indeed, as the Referee pointed out, we have not derived the answer for XY model but rather used known results in the literature. We have rephrased some sentences in the introduction to make it clear. We have, however, performed the summation over the effective form factors and derived the Fredholm determinant for the corresponding series. Equating these expressions allowed us to extract the dressed phase shift and use the effective form factors to derive the asymptotics. So, indeed, we can claim the method to derive asymptotics for the already known Fredholm determinant of a special kind. We hope that after the adjustments, this message is clear.

As for other remarks:

  • The neglected piece in the Fredholm det is said to vanish exponentially with x. But at finite temperature, so do the correlations. It should complicate computation of the next orders.

Indeed, sometimes even discarding this term is tricky as is discussed in Sec. 3.4. As a rule of thumb, we found out that these terms can be safely neglected if they vanish faster than the total correlator. However, we still feel that the next order computations are feasible once the estimates for the resolvent are obtained.

  • The calculation of the τ function was done by Korepin and Slavnov (Commun Math Phys 129, 103–113(1990)) and should be cited.

We have added the citation.

---

## Round 3 · Referee Report · Anonymous · 2021-3-2

Report

The authors consider the computation of form factors in non-interacting fermionic chains. Although these models have a long history, and many quantities can be computed analytically, some calculations remain difficult, especially pertaining to finite-entropy states beyond thermal equilibrium (GGEs).

As a main contribution, the authors develop an heuristic approach to address correlation functions in finite-entropy states, and recover exact results for spin-spin correlations in the XY model. More specifically, they introduce effective form factors for the fermions that absorb information about the state, significantly simplifying combinatorics of excitations, and making it as simple as the zero-temperature case.

The paper is well written and clear. The approach is scientifically rigorous, and the calculations presented appear to be correct. Furthermore, to my knowledge the idea presented is new, and could be potentially used in other contexts.

I don't have any particular comment regarding the content and the presentation, and overall I think the draft is already adequate for publication.

As my only comment, I would suggest the authors to expand a bit the discussion pertaining to the connections with some works in the literature of quantum quenches, which provides a strong motivation for this work, as the authors also write. For instance, in the conclusions, I think it would be appropriate to also mention quenches in the Lieb-Liniger model in the limit c->\infty, which in many ways is similar to XX. It seems to me that similar techniques based on form factors could apply in this case. Is this correct? In particular, relevant computations for some dynamical correlations functions were reported in

M. Kormos, M. Collura, and P. Calabrese, Analytic Results for a Quantum Quench from Free to Hard-Core One-Dimensional Bosons, Phys. Rev. A 89, 013609 (2014)

Later, these calculations were generalized to the Green function in

J. De Nardis and J.-S. Caux, Analytical Expression for a Post-Quench Time Evolution of the One-Body Density Matrix of One-Dimensional Hard-Core Bosons, J. Stat. Mech. 2014, P12012 (2014).

and finally extended to the case of 1D Lieb-Liniger anyons in

L. Piroli and P. Calabrese, Exact Dynamics Following an Interaction Quench in a One-Dimensional Anyonic Gas, Phys. Rev. A 96, 023611 (2017).

The authors could mention these results in the conclusions to better connect with the recent literature of quenches.

  • validity: -
  • significance: -
  • originality: -
  • clarity: -
  • formatting: -
  • grammar: -

Author:  Oleksandr Gamayun  on 2021-03-11  [id 1297]

(in reply to Report 3 on 2021-03-02)

We thank the Referee for their assessment of our work and for bringing attention to important papers. We have included them in the list of references.

---

## Round 4 · Author Response

Dear Editor,
we have implemented changes asked by the Referees and hereby submit the revised manuscript.
Best Regards,
The Authors
we have implemented changes asked by the Referees and hereby submit the revised manuscript.
Best Regards,
The Authors

---

## Round 4 · List of Changes

- Eq. (17): the index $\nu$ is inserted,
- Ref. Commun Math Phys 129, 103–113(1990)) is added in Introduction and before Eq. (11).
- A comment added after Eq. (8)
- The 5th paragraph in the introduction is rewritten.
- Small typos after Eqs. (59) and (63) are fixed
- As the third Referee requested we have added Refs. [79], [80] and [86]
- We have modified discussion after Eq. (53) and added Ref. [69]
- Citation [57] is added

---

## Editorial Decision

published